# Retrieval and validation of MetOp/IASI methane

Evelyn De Wachter[1], Nicolas Kumps[1], Ann Carine Vandaele[1], Bavo Langerock[1], and Martine De Mazière[1]

[1]Royal Belgian Institute for Space Aeronomy (BIRA-IASB), 1180 Brussels, Belgium

*Correspondence to:* Evelyn De Wachter (evelyn.dewachter@aeronomie.be)

**Abstract.** A new IASI methane product developed at the Royal Belgian Institute for Space Aeronomy (BIRA-IASB) is presented. The retrievals are performed with the ASIMUT-ALVL software based on the Optimal Estimation Method (OEM). This paper gives an overview of the forward model and retrieval concept. The usefulness of reconstructed Principal Component Compressed (PCC) radiances is highlighted. The information content study carried out in this paper shows that most IASI pixels contain between 0.9 and 1.6 independent pieces of information about the vertical distribution of $CH_4$, with a good sensitivity in the mid to upper troposphere. A detailed error analysis was performed. The total uncertainty is estimated to be 3.73% for a $CH_4$ partial column between 4-17 km. An extended validation with ground-based $CH_4$ observations at 10 locations was carried out. IASI $CH_4$ partial columns are found to correlate well with the ground-based data for 6 out of the 10 Fourier Transform Infrared (FTIR) stations with correlation coefficients between 0.60 and 0.84. Relative mean differences between IASI and FTIR $CH_4$ range between -2.31 and 4.04% and are within the systematic uncertainty. For 6 out of the 10 stations the relative mean differences are smaller than $\pm 1\%$. The standard deviation of the difference lies between 1.76 to 2.97% for all the stations.

## 1 Introduction

There is now a widespread scientific consensus on the profound influence of human activity on the global climatic system, particularly through increased emissions of greenhouse gases like carbon dioxide ($CO_2$) and methane ($CH_4$) since the pre-industrial era (Jardine et al., 2009). Although $CH_4$ is roughly 200 times less abundant in the atmosphere than $CO_2$, it is a more potent greenhouse gas. The comparative impact of $CH_4$ on climate change is more than 86 times greater than $CO_2$ over a 20-year period (Myhre et al., 2013). Identified $CH_4$ emission sources are either of biogenic, pyrogenic or thermogenic origin. $CH_4$ emissions of biogenic origin are related to anaerobic decomposition; a collection of processes by which microorganisms break down organic matter in the absence of oxygen. Examples are natural wetlands, oxygen-poor freshwater reservoirs, digestive systems of ruminants, rice paddies and waste treatment (Kirschke et al., 2013). Pyrogenic $CH_4$ is produced by the incomplete combustion of biomass and soil carbon during wildfires and of biofuels and fossil fuels. Thermogenic sources comprise the exploitation of oil, natural gas and coal and the natural degassing from the subsurface such as terrestrial seeps, marine seeps and mud volcanoes (Kirschke et al., 2013). The primary sink for atmospheric $CH_4$ is oxidation by hydroxyl radicals (OH), mostly in the troposphere, which accounts for about 90% of the global $CH_4$ sink (Kirschke et al., 2013). In addition, $CH_4$ is depleted at the surface by consumption by soil bacteria and by its reaction with chlorine radicals in the marine boundary layer.

These processes amount to a lifetime of atmospheric $CH_4$ of $\sim$9 years.

Since 2014, atmospheric $CH_4$ concentrations are rising faster than at any time in the past two decades and its concentration is now approaching the most greenhouse gas intensive Representative Concentration Pathway (RCP) trajectories (Saunois et al., 2016), the scenario pathways which were introduced by the Intergovernmental Panel on Climate Change (IPCC) in its fifth Assessment Report (AR5) in 2014. Its concentration has more than doubled since the pre-industrial period, reaching a new high of 1845$\pm$2 ppbv in 2015, an increase of 11 ppbv with respect to the previous year, as shown by the latest analysis of observations from the World Meteorological Organization (WMO) Global Atmosphere Watch (GAW) Programme (WMO, 2016). $CH_4$ is a challenging atmospheric component to study as its non-monotonous changes in the last decades and its interannual variability remain not fully understood (Nisbet et al., 2014). The mean annual growth rate of $CH_4$ decreased from 14 ppbv/yr in 1984 to near zero in 1999 (Dlugokencky et al., 2003). From 1999 to 2006, globally averaged $CH_4$ was relatively constant but atmospheric methane concentrations started rising again in 2007 with a global average growth of $\sim$6 ppbv/year (Nisbet et al., 2014). Latest analysis by Saunois et al. (2016) suggests that the cause of the atmospheric growth trend of the past decade is predominantly biogenic - most likely from agriculture - with smaller contributions from fossil fuel use and possibly wetlands.

Our current understanding of the natural and anthropogenic emissions of $CH_4$ is insufficient. Although the global OH sink of $CH_4$ and the sum of $CH_4$ sources is relatively well known, there are still large uncertainties about each of the individual sources of $CH_4$. However, due to its relatively short lifetime, it is now recognized that one of the most efficient methods to mitigate warming due to greenhouse gases on decadal time frames is to cut $CH_4$ emissions (Shindell et al., 2012). Global monitoring of $CH_4$ is therefore essential to increase our knowledge on how the different sources and sinks influence the atmospheric abundance of methane.

Atmospheric $CH_4$ has been measured continuously from space since 2003. Jacob et al. (2016) gives an extensive overview of past and future satellite missions dedicated to detect methane. Atmospheric $CH_4$ is detectable by its absorption of radiation in the shortwave infrared (SWIR) and thermal infrared (TIR). SWIR instruments, such as SCIAMACHY (Frankenberg et al., 2006), TANSO-FTS (Kuze et al., 2016) and the soon to be launched TROPOMI instrument (Hu et al., 2016), measure the solar radiation backscattered by the Earth and the atmosphere, and give a total atmospheric column of $CH_4$ with near uniform sensitivity in the troposphere (Jacob et al., 2016). TIR instruments measure the thermal radiation emitted by the Earth and the atmosphere, and operate in a nadir, limb or solar occultation observing mode. Limb and solar occultation detect $CH_4$ vertical profiles in the stratosphere and upper troposphere (Jacob et al., 2016). TIR nadir measurements provide integrated $CH_4$ columns in the middle to upper troposphere and allow day and nighttime concentrations, over land and sea. Examples are the AIRS instrument onboard the NASA Aqua satellite which has been providing global methane observations since 2002 (Xiong et al., 2008), TES which was operational from 2004 to 2011 (Worden et al., 2012), and IASI, launched onboard MetOp-A in October 2006 and on MetOp-B in September 2012 (Razavi et al., 2009; Crevoisier et al., 2009; Xiong et al., 2013; Siddans et al., 2016; García et al., 2017). With the launch of MERLIN foreseen in 2021, for the first time, active measurements will be made from space with an IPDA (Integrated Path Differential Absorption) lidar (Light Detecting And Ranging), which will provide atmospheric methane columns with high precision and unprecedented accuracy on a global scale (Pierangelo et al.,

2016).

As mentioned in the previous paragraph, in addition to the IASI $CH_4$ product presented here, other IASI $CH_4$ products exist. Crevoisier et al. (2009) uses a non-linear inference scheme based on neural networks, to derive a mid-tropospheric $CH_4$ column with peak sensitivity at about 230 hPa ($\sim$11 km), half the peak sensitivity at 100 and 500 hPa ($\sim$6 and 16 km), and no sensitivity to the surface. This dataset was previously only available for the tropical region between 30°S and 30°N but got extended to higher latitudes and is available through the Climate Change Initiative-Greenhouse Gas (CCI-GHG) project. The retrieval schemes of Siddans et al. (2016) and García et al. (2017) are based on the optimal estimation method (OEM), like the BIRA-IASB product. Different constraint matrices are used by the 2 products. García et al. (2017) et al use a Tikhonov-Philips slope constraint with strong regularisation (almost equivalent to a scaling retrieval). Siddans et al. (2016) use an a priori covariance matrix which describes the presumed errors in the a priori estimate of $CH_4$. The IASI $CH_4$ product presented in this paper follows a similar approach as Siddans et al. (2016).

In this paper, we present a new $CH_4$ product retrieved from IASI radiances with the ASIMUT-ALVL software developed at BIRA-IASB. This product complements the BIRA-IASB height-resolved IASI aerosol dust product (Vandenbussche et al., 2013). Sect. 2 introduces the IASI mission. In Sect. 3 we describe the IASI $CH_4$ radiative transfer and the retrieval setup, and the use of the Principal Component Compressed IASI spectra is addressed. The information content of the IASI $CH_4$ product is presented in Sect. 4. In addition, global distributions are shown and retrieval processing details are briefly discussed. In Sect. 5 the IASI $CH_4$ product is compared to ground-based measurements, providing an quality assessment of the retrieved BIRA-IASB $CH_4$ columns. The final section summarizes the main results of this work and discusses future work.

## 2 IASI

The Infrared Atmospheric Sounding Interferometer (IASI) onboard MetOp-A is a thermal cross-nadir scanning infrared sounder. Launched in October 2006, it is the first in a series of three, together programmed to provide measurements for a period of 15 years. The second instrument onboard MetOp-B was launched in September 2012 and the launch of MetOp-C is scheduled for October 2018.

IASI is a Fourier Transform Infrared (FTIR) spectrometer which measures the TIR radiation emitted by the Earth and the atmosphere. With a wide swath width of 2 x 1100 km it provides near-global coverage twice a day, with a local overpass time at $\sim$9:30 AM and PM. It has an instantaneous field of view (FOV) at nadir with a spatial resolution of 50 km x 50 km, composed of 2 x 2 circular pixels, each corresponding to a 12 km diameter footprint on the ground at nadir (Clerbaux et al., 2009). IASI has three spectral bands in the spectral range from 645 to 2760 $cm^{-1}$ (3.62 to 15.5 µm), provided as a continuous spectrum with an apodized spectral resolution of 0.5 $cm^{-1}$ and spectral sampling of 0.25 $cm^{-1}$.

Designed to provide highly accurate temperature and humidity profiles for numerical weather prediction, the IASI mission allows simultaneous global observations of the air composition with an excellent spatial resolution. From the atmospheric spectra recorded by the instrument, concentrations of several trace gases can be monitored, enhanced levels of pollution can be

detected, and particle types can be determined to some extent. In the longer term the continuity of the programme is ensured with the IASI-NG mission that will extend the IASI observations for 15-20 more years (Clerbaux et al., 2016).

## 3 The IASI CH$_4$ retrieval method

The IASI CH$_4$ profiles are retrieved with the ASIMUT-ALVL software developed at BIRA-IASB (Vandaele et al., 2006).
ASIMUT-ALVL is modular software for radiative transfer (RT) calculations and inversions in planetary atmospheres. The code has been developed with the objective to be as general as possible, accepting different instrument types and different geometries. ASIMUT-ALVL has been coupled to the SPHER/TMATRIX (Mishchenko and Travis, 1998) and LIDORT (Spurr, 2006) codes to include the complete treatment of the scattering effects into the RT calculations. It has a specific interface dealing with the IASI instrument characteristics and IASI input information and is also used for the IASI aerosol dust retrievals (Vandenbussche et al., 2013). The RT simulations are performed with the ASIMUT-ALVL RT code for the IASI CH$_4$ data product while the LIDORT RT code is used for the IASI aerosol dust retrievals in order to include all scattering effects due to aerosols. Both IASI retrieval products use the same retrieval module, based on the formalism of the OEM (Rodgers, 2000). Initially developed for Earth observation missions, ASIMUT-ALVL has also been adapted for planetary atmospheres, in particular those of Venus (Vandaele et al., 2008) and Mars (Drummond et al., 2011) and is now the reference code for the NOMAD instrument onboard ExoMars TGO (Robert et al., 2016).

### 3.1 Forward model

The ASIMUT-ALVL RT module simulates atmospheric transmittances and radiances for cases under local thermodynamical equilibrium and where scattering can be neglected. A detailed description of the radiative transfer model is given in Vandaele et al. (2006). The spectral range considered for the CH$_4$ retrieval is the 1210-1290 cm$^{-1}$ region covering part of the ν4 spectral band. EUMETSAT IASI L2 skin temperature (T$_{skin}$), temperature and water vapour profiles are used as input for the radiative transfer calculations. The spectroscopic parameters for CH$_4$, N$_2$O and other species are taken from the HITRAN 2012 database (Rothman et al., 2013). The IASI Instrument Line Shape (ILS) is characterized by a Gaussian function with a 0.5 cm$^{-1}$ FWHM. Frequency dependent emissivity maps are provided by Zhou et al. (2011). Fig. 1 shows an example of measured and simulated radiances in the 1210-1290 cm$^{-1}$ spectral region and the residual (difference between measured and simulated radiances). In the lower panel, the overlapping contributions of the different molecules CH$_4$, H$_2$O and N$_2$O are illustrated. Here the radiances are simulated under the assumption of a single-species atmosphere containing either CH$_4$, H$_2$O or N$_2$O. The top panel shows a negligible bias and a 1-σ standard deviation comparable to the radiometric noise of 2x10$^{-8}$ W/(cm$^2$ sr cm$^{-1}$) (see Sect. 3.2). Certain spectral ranges in the considered spectral region are not well simulated by the radiative transfer model, leading to outliers in the residuals with absolute differences larger than 5x10$^{-8}$ W/(cm$^2$ sr cm$^{-1}$), for example at 1246 cm$^{-1}$ and 1252 cm$^{-1}$. These spectral ranges are masked in the retrieval set-up, i.e. the signal-to-noise is set to zero at these spectral points, so that no information is lost.

Only IASI L1C spectra with a cloud fraction $< 10\%$ based on the EUMETSAT IASI L2 fractional cloud cover product are processed.

## 3.2 Retrieval and error characterization

The ASIMUT retrieval module is based on the OEM (Rodgers, 2000) where the Jacobians are calculated analytically. The characteristics of the IASI retrieval are summarized in Table 1. The state vector includes $T_{skin}$, 23-level $CH_4$, $N_2O$ and $H_2O$ profiles and a $CO_2$ total column. The $T_{skin}$ a priori is taken from the EUMETSAT IASI L2 $T_{skin}$ product. The a priori profiles $x_a$ and covariance matrices $S_a$ for $CH_4$ and $N_2O$ are based on a climatology from the WACCM model. A single global $CH_4$ $x_a$ profile is used for all the retrievals, representative of a mid-latitude $CH_4$ profile. Therefore the atmospheric $CH_4$ variations observed are a results of the variability of atmospheric $CH_4$ rather than the a priori information. The $CH_4$ covariance matrix represents the highest variability at the surface and in the upper troposphere-lower stratosphere (UTLS). The variability in the UTLS is representative for the variability of the $CH_4$ gradient at the tropopause which is different at different latitudes. The $H_2O$ a priori uncertainty covariance matrix is characterised by an uncertainty covariance matrix with a 10% standard deviation on the diagonal and an exponential decaying correlation width of 6 km. The EUMETSAT IASI L2 water vapour profile is used as the $H_2O$ a priori profile $x_a$. The interfering species $HNO_3$ and $O_3$ are included in the RT calculations, their a priori values are provided by the WACCM model.

A diagonal measurement uncertainty covariance $S_e$ is taken, with the radiometric noise set to $2x10^{-8}$ W/(cm$^2$ sr cm$^{-1}$). This value is conservative, about a factor 5 higher than the estimated radiometric noise in this spectral region of $4x10^{-9}$ W/(cm$^2$ sr cm$^{-1}$) (Clerbaux et al., 2009). It includes not only the measurement uncertainty, but also the uncertainties in the temperature and water vapour profile, the spectroscopic parameters and surface emissivity (De Wachter et al., 2012).

Fig. 2 presents the $CH_4$ a priori profile (pink) and retrieved $CH_4$ profile (blue) in volume mixing ratio (vmr) for a pixel at 27°N. The horizontal bars represent the retrieval uncertainty. The averaging kernel (AK) is given in the middle figure. It shows that the sensitivity of the IASI $CH_4$ product lies in the 800-100 hPa ($\sim$2-16 km) range. The right plot of Fig. 2 displays the vertical profiles of the retrieval uncertainties together with the $CH_4$ a priori variability (black line). The square root of the diagonal elements of the uncertainty is plotted. The $CH_4$ a priori variability is calculated from the square root of the diagonal of the a priori uncertainty covariance matrix. The retrieval is quite constrained with an a priori variability of a few percent at the surface going up to 7-8% at 20 km. Following Rodgers (2000) the error sources contributing to the total retrieval uncertainty are 1) the smoothing error, which accounts for the vertical resolution of the retrieved $CH_4$, 2) the error due to uncertainties in forward model parameters such as spectroscopy, the temperature profile, surface emissivity and 3) the IASI measurement uncertainty. For IASI the forward model uncertainties are included in the measurement uncertainty. As we can see from Fig. 2, the dominant source of uncertainty is the smoothing uncertainty. The total retrieval uncertainty declines from 3% at the surface to $\sim$2% between 800 and 200 hPa, the altitude range of maximum sensitivity. Above 200 hPa the total retrieval uncertainty increases rapidly up to $\sim$4% at 100 hPa and $\sim$6% at 60 hPa.

### 3.3 Retrievals with PCC L1C data

The CH$_4$ profiles are retrieved from IASI radiances recomposed from the EUMETSAT Principal Component Compressed (PCC) L1C dataset (Hultberg, 2009). The use of PCC data allows both noise filtering and a large reduction in data volume compared to the use of raw radiances. Our main motivation is the large reduction in data storage. One year of the original IASI L1C (BUFR format) data amounts to 10 Tb which is reduced to 1 Tb for the PCC data. Fig. 3 shows the raw and PCC radiances for a random pixel in the CH$_4$ $\nu 4$ spectral band. Differences between raw radiances and PCC radiances lie in the IASI radiometric noise level ($4 \times 10^{-9}$ W/(cm$^2$ sr cm$^{-1}$)) as given by the IASI radiometric noise figure from Clerbaux et al. (2009). This is a factor 5 lower than the conservative radiometric noise level of $2 \times 10^{-8}$ W/(cm$^2$ sr cm$^{-1}$) used in the CH$_4$ retrieval (see Sect. 3.1). Fig. 4 compares the CH$_4$ concentrations retrieved with the PCC L1C data with those retrieved with the raw radiances for March 2011 and September 2013 for daytime and nighttime retrievals between 60°S and 70°N. We find an excellent correlation (R=1) between the retrieved concentrations and negligible biases of 0.0026% and 0.025% with a 1-$\sigma$ standard deviation of $\leqslant$0.12%. With these results we are confident to use the PCC-reconstructed radiances.

## 4 The BIRA-IASB IASI CH$_4$ product

### 4.1 Information content

For correct interpretation of the data one needs to consider the vertical sensitivity of the retrieved CH$_4$ profile. This information is contained in the averaging kernel (AK), which is provided with each retrieved CH$_4$ profile. The peak of each AK gives the altitude of maximum sensitivity. Its full width at half maximum can be interpreted as the vertical resolution of the retrieval. Averaging kernels are variable, as can be seen from Fig. 5. Given is the CH$_4$ AK for 3 pixels in March 2013 at 3 different geographical locations; at northern mid-latitudes (52 °N), in the tropics (4 °N) and at southern mid-latitudes (47 °S). In the tropics, the CH$_4$ sensitivity lies in the 850-100 hPa ($\sim$1.5-16 km) range, at mid-latitudes, in the 700-200 hPa ($\sim$3-12 km) range. For the 3 geographical locations, the sensitivity is reduced in the boundary layer, which is typical for thermal infrared sounders. In each figure the Degree of Freedom for Signal (DOFS) is given, which is an estimate of the number of independent pieces of information contained in the measurement. It is the trace of the AK. One independent piece of information (1.07 < DOFS < 1.45) is deduced for the 3 geographical locations. Maps of the CH$_4$ DOFS for February and August 2013 are presented in Fig. 6. DOFS values for daytime retrievals are shown in the 2 lefthand figures, DOFS values for nighttime retrievals are shown in the 2 righthand figures. For both seasons and day- or nighttime retrievals we typically have DOFS values in the tropics around 1.4. For the Northern Hemisphere (NH), at mid-latitudes, we see higher DOFS values in August (NH summer) than in February (NH winter). In February values can become less than 1 for latitudes > 40°N. The variability of the AK and hence the DOFS is dependent on the thermal contrast (the difference between the surface temperature and the temperature of the first atmospheric vertical layer), which exhibits significant geographical, seasonal, and diurnal variability. The retrieval sensitivity is favourable, and hence the DOFS is high, when thermal contrast is high. In general the thermal contrast or the DOFS is higher during the day, over land, and over dry, sparsely vegetated regions (Clerbaux et al., 2009). This pattern is visible in Fig. 6, where DOFS

values are generally higher for daytime retrievals compared to the nighttime retrievals, and where high DOFS values are found for the daytime observations at desert regions of Africa and Australia. The DOFS for different latitudinal bands for February and August 2013 is presented in Fig. 7. DOFS values for daytime retrievals are provided in the 2 lefthand figures, DOFS values for nighttime retrievals are shown in the 2 righthand figures. This figure confirms that DOFS values in the tropics are typically

around 1.4. For August (NH summer), on the global scale, the DOFS values for daytime retrievals range between 1 and 1.8 (between 0.9 and 1.6 for nighttime retrievals). For February (NH winter), values range between 0.4 and 1.7 (between 0.4 and 1.6 for nighttime retrievals), when values can become less than 1 for latitudes $> 40°$N. So overall one independent piece of information is retrieved with a good sensitivity in the mid to upper troposphere.

## 4.2    Global distributions

Monthly mean global daytime distributions for the year 2013 are presented in Fig. 8. IASI $CH_4$ partial columns between 4 and 17 km between 60°S and 70°N are shown for the IASI morning overpass. $CH_4$ concentrations are averaged over the four, 2 x 2 circular IASI pixels, which are measured simultaneously (see Sect. 2) and binned on a 1° x 1° grid. Areas with missing data correspond to areas which were identified as cloudy by the EUMETSAT IASI L2 fractional cloud cover product, or correspond to areas where not all of the 4 simultaneously measured pixels converged in the retrieval.

We see a latitudinal gradient with higher concentrations in the Northern Hemisphere (NH) than in the Southern Hemisphere (SH), which is consistent with the fact that most of the methane sources are located in the Northern Hemisphere. In the NH, higher $CH_4$ concentrations are found during boreal summer than during boreal winter. This summer increase of mid to upper tropospheric $CH_4$ has also been observed by AIRS (Xiong et al., 2010). Van Weele et al. (2011) examined the $CH_4$ variability in the upper troposphere and lower stratosphere between ∼6-25 km using aircraft observations and the TM5-chem-v3.0 chemistry

transport model (Krol et al., 2005; Huijnen et al., 2010). They also found higher $CH_4$ mixing ratios at the 500 hPa level (∼6 km) at high latitudes during boreal summer compared to winter concentrations and attributed the winter minimum to enhanced downward transport from the stratosphere.

Methane observed in the boundary layer by surface stations from the NOAA network displays a reversed seasonal cycle in the NH (Dlugokencky et al., 2009). These results demonstrate the added value of thermal infrared $CH_4$ measurements which have

a sensitivity at higher altitudes.

## 4.3    Error analysis

In Sect. 3.2 we discussed the two error sources which contribute to the total retrieval error: the smoothing error, which accounts for the low vertical resolution of the retrievals, and the measurement error. Their uncertainties are estimated following Rodgers (2000) and are shown in Fig. 2. Additional sources of error propagating into the total retrieval error are due to uncertainties in

forward model parameters or ancillary data used in the inversions. These error sources are currently not explicitly taken into account in the ASIMUT retrieval software. We therefore estimated the uncertainties by forward model parameters or ancillary data by a perturbation method, following Barret et al. (2002, 2003). A set of spectra in a latitude-longitude band between 60°S-70°N and 120-125°E were selected, comprising a set of 4000 spectra. This is a representative set for the latitudinal

coverage of the IASI $CH_4$ dataset, with spectra over land and water. For the different error sources considered, IASI $CH_4$ is retrieved for this set of spectra with the original set-up and the uncertainty added to the specific error source. The uncertainty of this error source on the IASI $CH_4$ partial column is then estimated as the difference between the newly retrieved IASI $CH_4$ partial column (with the uncertainty of the error source added) and the IASI $CH_4$ partial column from the current optimized retrieval set-up. The different error sources and their uncertainties are listed in Table 2, as well as the results of the estimated uncertainties of the IASI $CH_4$ 4-17 km partial column for each individual error source.

The uncertainty of the temperature profile on the $CH_4$ partial column is estimated by substituting the IASI L2 temperature profiles with the ECMWF ERA-Interim (Dee et al., 2011) re-analysis temperature profiles. ECMWF ERA-Interim re-analysis data is available at 6 hourly intervals with a horizontal resolution of $\sim 0.75°$ in latitude and longitude. The temperature profiles are interpolated to the location and time of the IASI pixel and the retrieved $CH_4$ is compared with the $CH_4$ partial column of the optimized retrieval set-up. For the $CH_4$ absorption lines the uncertainty on the line intensity, the air and self broadening coefficients is set to 2%. This is consistent with what García et al. (2017) considered in their uncertainty estimation. For the interfering species, $N_2O$, $H_2O$ and isotopologues, which are simultaneously retrieved, we also set the uncertainties on the spectroscopic parameters (line intensity and air and self broadening coefficients) to 2%. We also estimated a systematic uncertainty of the IASI $CH_4$ a priori of 2%. Following García et al. (2017) the uncertainty on the emissivity is 1% for all wavenumbers. For the PCC uncertainty we calculated the difference between IASI $CH_4$ retrieved from PCC spectra and from raw spectra, as already shown in Sect. 3.3. The smoothing and measurement uncertainty are estimated as in Sect. 3.2.

The third column in Table 2 lists the results. The dominant sources of error are the smoothing error and the $CH_4$ line intensity with an uncertainty on the IASI $CH_4$ 4-17 km partial column of 2.45% and 1.93% respectively. Other error sources contributing significantly to the uncertainty of the $CH_4$ 4-17 km partial column are the temperature profile (1.40%), the $CH_4$ broadening coefficients (1.09%) and the measurement uncertainty (0.95%). There is also a non-negligible contribution of the emissivity uncertainty of 0.27%. Uncertainties in the spectroscopic parameter of $N_2O$, $H_2O$ and its isotopologues do not significantly contribute to the uncertainty in $CH_4$. The systematic uncertainty of the IASI $CH_4$ a priori also has a negligible effect of 0.06%. Combining the different contributions to the IASI $CH_4$ error budget, we estimated a total uncertainty on the $CH_4$ 4-17 km partial column of 3.73%. If we consider the temperature, measurement, PCC reconstruction and smoothing uncertainty as random error sources we get an estimate of the precision of the IASI $CH_4$ 4-17 km partial column of 2.98%. If we consider uncertainties in the spectroscopy and emissivity as systematic error sources, the systematic uncertainty of the $CH_4$ 4-17 km partial column is 2.23%.

## 4.4 Retrieval output and processing

The BIRA-IASB IASI $CH_4$ product is delivered in HDF5 format. Daily daytime and nighttime observations are provided in separate files. The HDF files contain $CH_4$ profiles, the retrieval uncertainty, the $CH_4$ a priori profiles and averaging kernels. BIRA-IASB entered Phase 2 of the CCI-GHG project (Buchwitz et al., 2015) and a dataset has been generated for the years 2011-2014, contributing to Climate Research Data Package No.4 (CRDP#4). These retrievals were performed between 60°S and 70°N and can be downloaded from http://iasi.aeronomie.be/. Data is processed on a High Performance computing system

(HPC) with 2 x 55 nodes of 24 Central Processing Units (CPUs) where the user obeys a quota-based use. One day of IASI $CH_4$ data (for the 60°S and 70°N region) is processed in 48 hours on 1 node with 24 CPUs.

## 5    Validation

Ground-based data was collected from 10 FTIR stations from the Network for the Detection of Atmospheric Composition

Change (NDACC). The stations chosen are operated on a quasi-continuous basis and deliver $CH_4$ vertical profiles. Certain stations provide limited observations since they only recently entered the NDACC network or since they only make campaign measurements. We therefore excluded stations with fewer than 200 collocations due to insufficient collocation points for a statistically significant comparison. NDACC FTIR $CH_4$ profiles have good sensitivity in the troposphere and stratosphere with 2 to 3 independent pieces of information. Note, the NDACC $CH_4$ retrieval is not fully harmonized yet for all the NDACC

stations. This work is ongoing as part of the Horizon 2020 Gap Analysis for Integrated Atmospheric ECV CLImate Monitoring (GAIA-CLIM) project (http://www.gaia-clim.eu/).

We performed a detailed comparison between IASI and NDACC $CH_4$ partial columns between 4 and 17 km at these 10 NDACC stations for the period 2011 to 2014. Since the two retrievals have been computed with a different a priori, the NDACC retrieved profiles are adjusted for the comparison. Following Rodgers and Connor (2003) (equation 10), the term

$(\mathbf{A_{NDACC}} - \mathbf{I}) \cdot (\mathbf{x_{a,NDACC}} - \mathbf{x_{a,IASI}})$ is added to each NDACC retrieval to adjust for the different a priori profile used in the IASI retrieval. Here $\mathbf{A_{NDACC}}$ is the NDACC averaging kernel, $\mathbf{I}$ the unity matrix, $\mathbf{x_{a,NDACC}}$ the NDACC $CH_4$ a priori profile and $\mathbf{x_{a,IASI}}$ the IASI $CH_4$ a priori profile.

In addition, to account for the different resolution between the IASI and the higher resolved NDACC FTIR profiles, a smoothing is applied to the (a priori adjusted) NDACC profile $\mathbf{x_{NDACC}}$ by the IASI averaging kernel:

$$\hat{\mathbf{x}}_{\mathbf{NDACC}} = \mathbf{x_{a,IASI}} + \mathbf{A_{IASI}} \cdot (\mathbf{x_{NDACC}} - \mathbf{x_{a,IASI}}) \tag{1}$$

where $\hat{\mathbf{x}}_{\mathbf{NDACC}}$ is the smoothed or convolved NDACC $CH_4$ profile and $\mathbf{x_{a,IASI}}$ and $\mathbf{A_{IASI}}$ are the IASI a priori profile and averaging kernel.

From the IASI and smoothed NDACC $CH_4$ profiles partial columns are calculated between 4 and 17 km. The average is taken of IASI pixels selected within 3 hours of the NDACC FTIR measurement, in a 0.5°-1.5° latitude-longitude-box centred around

the point on the line of sight (LOS) where the FTIR measurement has maximum sensitivity (typically at 5km altitude on the LOS). To guarantee a certain homogeneity of the NDACC data with NDACC $CH_4$ profiles of comparable quality we applied a filtering on some of the NDACC data when large outliers where found. We also applied a filtering to the IASI $CH_4$ profiles. We omitted IASI pixels with DOFS<0.85 and when the root mean square of the residual>$2.2\text{x}10^{-8}$ W/(cm$^2$ sr cm$^{-1}$).

Fig. 9 summarizes the results in a bar chart giving the relative difference between IASI and smoothed NDACC partial columns

between 4 and 17 km at the different NDACC stations. The relative mean difference $\Delta$=(IASI-NDACC)/NDACC and standard deviation of the difference ($\sigma$) in percentage is given for each station. Relative mean differences between IASI and NDACC lie between -2.31 and 0.18% (of which 6 stations out of 10 less than ±1%) with exception of the Thule station, where IASI is

biased high with respect to NDACC by 4.04%. The standard deviation of the difference lies in the range 1.76 to 2.97% for the 10 stations.

It is important to compare these results with the uncertainty budget of the IASI and the NDACC $CH_4$ partial columns. As given by Rodgers and Connor (2003) (equation 30), $\mathbf{S_\Delta}$, the covariance of the difference IASI-NDACC, can be calculated as:

$$\mathbf{S_\Delta} = (\mathbf{A_{IASI}} - \mathbf{A_{IASI}A_{NDACC}})\mathbf{S_{a,IASI}}(\mathbf{A_{IASI}} - \mathbf{A_{IASI}A_{NDACC}})^T$$
$$+ \mathbf{S_{IASI}}$$
$$+ \mathbf{A_{IASI}S_{NDACC}A_{IASI}^T}$$

The first term is the smoothing uncertainty of the comparison ensemble (the smoothed and a priori-corrected NDACC and IASI product) with $\mathbf{S_{a,IASI}}$ the IASI a priori uncertainty covariance matrix. $\mathbf{S_{IASI}}$ is the IASI retrieval uncertainty covariance exluding the smoothing uncertainty and $\mathbf{S_{NDACC}}$ is the NDACC retrieval uncertainty covariance without the smoothing uncertainty. We compare the systematic and random uncertainty on the difference directly to the mean difference and standard deviation of the difference between IASI and NDACC. NDACC provides systematic and random uncertainty covariances for the different stations, with exception of Jungfraujoch. For IASI we set the random uncertainty $\mathbf{S_{a,IASI}^{rand}}$ equal to the IASI apriori uncertainty covariance matrix used in the IASI retrieval. We calculated a systematic component $\mathbf{S_{a,IASI}^{syst}}$ with a 2% standard deviation of the a priori profile values. Separating the systematic and random component of $\mathbf{S_{IASI}}$ is less straightforward. Here we only consider the IASI measurement uncertainty as the random uncertainty and we do not consider the systematic component.

Table 3 lists the results of this analysis with the relative mean differences, the standard deviation of the differences and the mean values of the systematic and random uncertainties. We have a good agreement where the IASI-NDACC mean differences lie within the systematic uncertainty. Also for Thule the mean difference of 4.04% is within the systematic uncertainty of 5.28%. The standard deviation of the difference is within the random uncertainty for all stations. We did notice a current underestimation of the random uncertainty of the NDACC $CH_4$ retrievals. They were found to be less than 1% for 6 out of the 10 NDACC stations. Also note the spread in uncertainty estimates, especially for the systematic component. This is due to the differences in reported systematic and random error covariances from the different NDACC stations. The ongoing work in the GAIA-CLIM project will harmonize the error characterization for all NDACC stations in the coming period. This comparison stresses the importance of this harmonization work.

Scatter plots of collocated partial columns are presented in Fig. 10. We find good correlations (R = 0.67-0.84) for the **high-latitude** stations Eureka, Thule and Kiruna. Good correlations are found as well for the **mid-latitude** stations Jungfraujoch (R=0.81) and Zugspitze (R=0.68), while the mid-latitude station Toronto performs poorer with a correlation of 0.52. The **tropical** island stations Izaña, Maido and Mauna Loa show poor correlations (R = 0.15-0.37) although biases are below 1.20% for these stations. For the most Southern station Wollongong (34°S) we find a correlation of 0.60. Several tests were performed to explain the poorer correlations found at the tropical island stations. We applied a stronger filtering on the IASI and NDACC data but found no improvement. We investigated a possible relation of IASI land or IASI sea pixels with differences between the IASI and NDACC retrieved $CH_4$ but found no correlation. We therefore attribute the poorer correlations at the Izaña, Mauna

Loa and Maido stations to the lower $CH_4$ variability we see at these locations compared to the other stations. In addition, at Maido and Mauna Loa, we see a few outliers which could explain the poorer linear regression fit at these stations.

## 6    Discussion, conclusion and outlook

Although $CH_4$ is a more effective greenhouse gas than $CO_2$, it has a much shorter atmospheric lifetime than $CO_2$ that can remain in the atmosphere for hundreds or thousands of years. Therefore the mitigation of $CH_4$ emissions provides an opportunity for alleviating climate change in the short-term future (Kirschke et al., 2013). Global monitoring of $CH_4$ is essential to study the evolution of atmospheric $CH_4$ and to help increase our knowledge on how the different sources and sinks influence its atmospheric abundance.

In this paper, we presented a new IASI $CH_4$ retrieval product developed at BIRA-IASB. Global distributions of $CH_4$ were derived from IASI radiances with the ASIMUT-ALVL software based on the OEM. A detailed description of the forward model, the retrieval strategy and the use of PCC L1C data was given. $CH_4$ concentrations retrieved from raw radiances and PCC-reconstructed radiances showed an excellent correlation and negligible mean differences of < 0.026% (< 0.46 ppbv).

We presented the latitudinal distribution of the DOFS for different seasons. We showed that, between 60°S and 70°N, the DOFS values range between 1 (0.9) and 1.8 (1.6) for daytime (nighttime) retrievals for NH summer. In NH winter values can become less than 1 for latitudes > 40°N. In tropical scenes DOFS values are typically around 1.4, with a good sensitivity in the mid to upper troposphere.

A quality assessment of the retrieved IASI $CH_4$ product was given by a detailed comparison with ground-based FTIR observations recorded at 10 NDACC stations. The BIRA-IASB product was compared to smoothed NDACC FTIR $CH_4$ partial columns between 4 and 17 km for the years 2011 to 2014. We found a very good agreement between both products with differences within the systematic uncertainty. Mean difference values range between -2.31 and 4.04% for the 10 stations. Absolute differences are less than 1% for 6 stations out of 10. The standard deviation of the difference lies in the range 1.76 to 2.97% for all the stations. These values are within the random uncertainty of IASI and NDACC. Very good correlations are found for 6 out of the 10 NDACC stations with correlation coefficients between 0.60 and 0.84. Particularly for the 3 high-latitude stations we find high correlations, as well as for the 2 high-quality mid-latitude stations Jungfraujoch and Zugspitze.

Sect. 1 highlighted the need to improve our current understanding of the global budget of $CH_4$. The Requirements Baseline Document (RBD) of the CCI-Climate Modelling User Group (CMUG) stipulates the observational requirements for regional source/sink determination of $CH_4$. The RBD states that a $CH_4$ profile/tropospheric $CH_4$ column observation at a horizontal resolution of 50 km requires a precision of 1% and an accuracy of 2% for a 6h observing cycle (CMUG-RBD, 2015). To reach these demanding requirements improvements in the precision of the IASI 4-17 km partial column are needed. The error analysis in Section 4.3 gives a random uncertainty of 2.98% with the largest contribution coming from the smoothing error and uncertainty of the temperature profile. The simultaneous retrieval of the temperature profile could improve the IASI precision and will be investigated in the near future. The systematic uncertainty estimate in Section 4.3 of 2.23% could be improved by reducing spectroscopic uncertainties. Continuous efforts will be made on improving the IASI $CH_4$ retrievals as to these issues

and enhancing their precision.

Future work will further focus on extending the validation with additional datasets. Validation measurements for atmospheric vertical profiles for $CH_4$ are limited and very diverse. An innovative atmospheric sampling system called AirCore (Karion et al., 2010; Membrive et al., 2017) has been demonstrated to be a reliable concept to make vertical profile measurements of

$CO_2$, $CH_4$ and CO from the surface up to $\sim$30 km. Although campaign-based, these high precision measurements provide a promising and novel validation tool. One of the next steps is to compare the IASI $CH_4$ with AirCore $CH_4$ profiles. Further, a global scale comparison with the neural network IASI-$CH_4$ product (Crevoisier et al., 2009) or with one of the new OEM IASI-$CH_4$ products recently published and under revision (Siddans et al., 2016; García et al., 2017) would be of particular interest.

IASI provides day- and nighttime measurements over land and sea and has a high spatial coverage. Its follow-up missions guarantee a long continuity of observations and its successor, the IASI-NG next-generation instrument, will ensure a continuity of data until after 2040. IASI-NG's spectral resolution and signal-to-noise ratio will be improved by a factor of two. It will fly on the three second-generation MetOp-SG-A series, scheduled to launch in 2021, 2028 and 2035. IASI provides therefore a great opportunity for continuous monitoring of the atmospheric composition on a fine spatio-temporal scale.

Furthermore, future work will focus on comparing the IASI concentrations with tagged simulations of $CH_4$ to see whether the model output is supported by the IASI data. With this research we want to provide a better understanding of the $CH_4$ budget, which can help target the pertinent sources for reducing $CH_4$ emissions and the associated climate impact of this greenhouse gas.

The BIRA-IASB IASI $CH_4$ dataset is available through the European Space Agency (ESA) CCI-GHG project and can be down-

loaded from http://iasi.aeronomie.be/. Data is available for the years 2011-2014 between 60°S and 70°N and $CH_4$ profiles, a priori profiles, retrieval uncertainties and averaging kernels are provided.

*Acknowledgements.* The IASI mission is a joint mission of Eumetsat and the Centre National d'Etudes Spatiales (CNES, France). The IASI L1C data are distributed in near real time by Eumetsat through the Eumetcast system distribution. This work was conducted as part of the IASI.flow (Infrared Atmospheric Sounding with IASI and Follow-on missions) project, funded by the Belgian Science Policy Office

and the European Space Agency (ESA-Prodex programme). Additional support was provided by the ESA CCI-GHG project through the Optional Workpackage 706 TIRS (CO2 and CH4 from Thermal Infrared Sounders: IASI and ACE-FTS). The ground-based data used in this publication were obtained as part of the Network for the Detection of Atmospheric Composition Change (NDACC) and are publicly available (see http://www.ndacc.org).

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

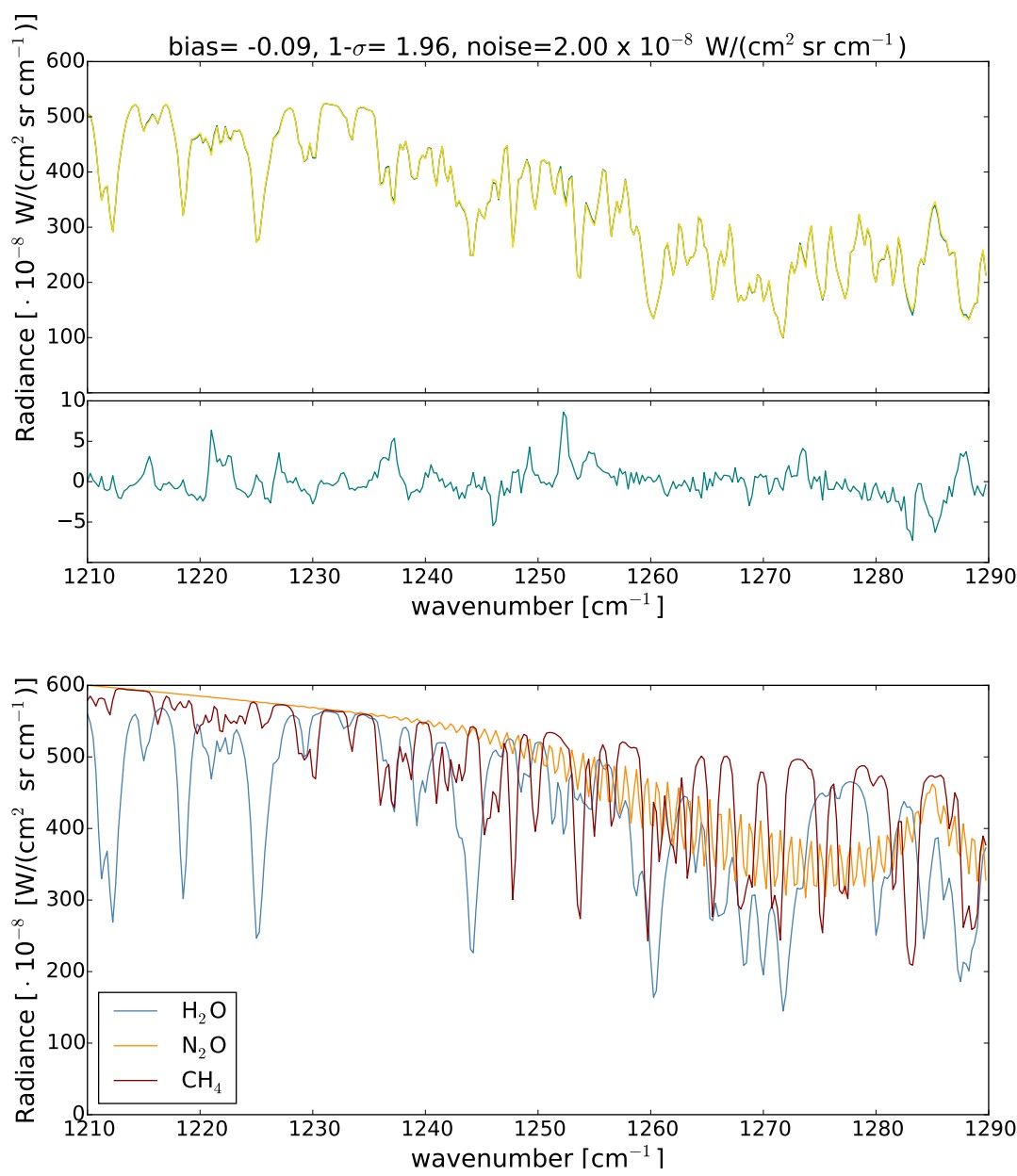

**Figure 1.** Top panel : [top] Measured (blue) and simulated (yellow) radiances. [bottom] Measured minus simulated radiances. The mean difference (bias), 1-σ standard deviation of the difference and radiometric noise-value used in the retrieval (all in x $10^{-8}$ W/(cm$^2$ sr cm$^{-1}$)) are given in the title. Bottom panel : Three simulated radiances under the assumption of a single-species atmosphere containing either only CH$_4$, H$_2$O or N$_2$O, showing the contribution of the different prominent molecules in this spectral region.

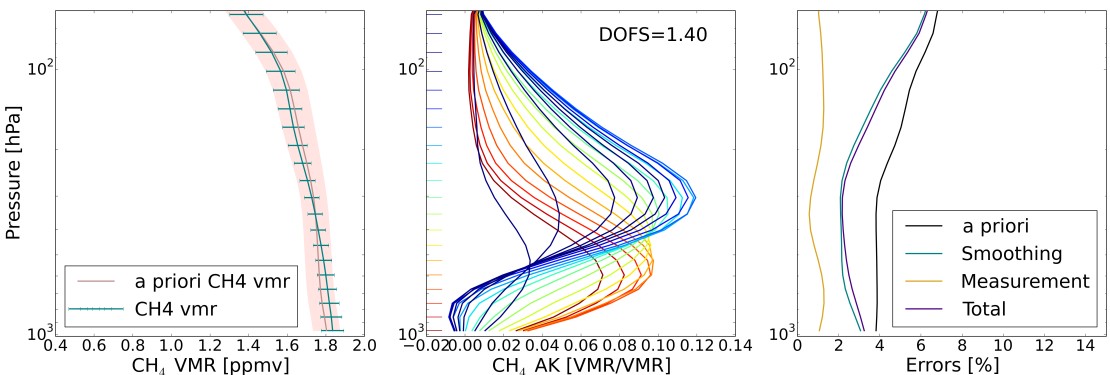

**Figure 2.** [left] Retrieved and a priori $CH_4$ vmr profile in ppmv for an observation on the 2$^{nd}$ of July 2013. The pink shaded area is the a priori variability and the horizontal blue bars are the retrieval uncertainty. [middle] Averaging kernel of the retrieval with a DOFS of 1.40. [right] $CH_4$ uncertainty profiles in percentage. Given are the measurement (yellow) and smoothing (blue) uncertainty which contribute to the total (purple) uncertainty. The black line represents the variability of the a priori as calculated from the square root of the diagonal elements of the a priori uncertainty covariance matrix $S_a$.

|  | CH$_4$ |
|---|---|
| Spectral range | 1210-1290 cm$^{-1}$ |
| State vector | CH$_4$, H$_2$O and N$_2$O profile, |
|  | CO$_2$ total column, T$_{skin}$ |
| Pressure, temperature, RH | IASI L2 |
| Spectroscopy | HITRAN 2012 |
| Emissivity | Zhou et al. (2011) |
| a priori information | WACCM + IASI L2 |

**Table 1.** Characteristics BIRA-IASB CH$_4$ retrieval.

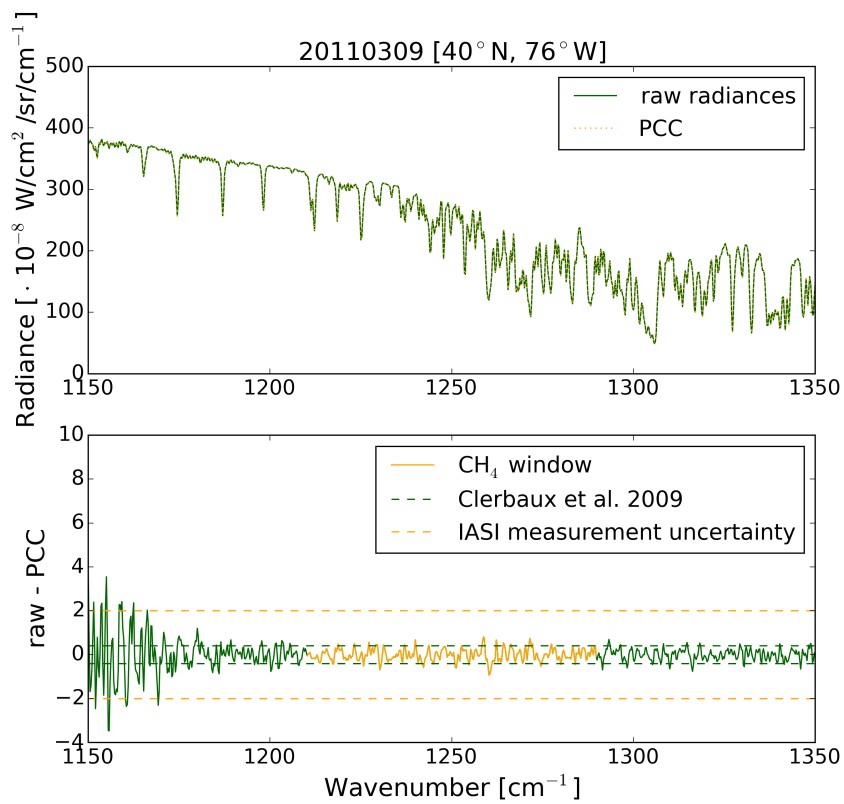

**Figure 3.** Radiances from a nighttime pixel on the 9[th] of March 2011 at 40°N and 76°W. [top] Raw radiances and PCC-reconstructed radiances. [bottom] Difference between raw radiances and PCC radiances. The CH$_4$ spectral retrieval window is highlighted in orange. The green horizontal dashed lines indicates the IASI radiometric noise at 1250 cm$^{-1}$ as given by Clerbaux et al. (2009) and the orange horizontal dashed lines indicate the IASI radiometric noise defined in the retrieval (see Sect. 3.1). The differences between the raw and PCC radiances are within the IASI radiometric noise.

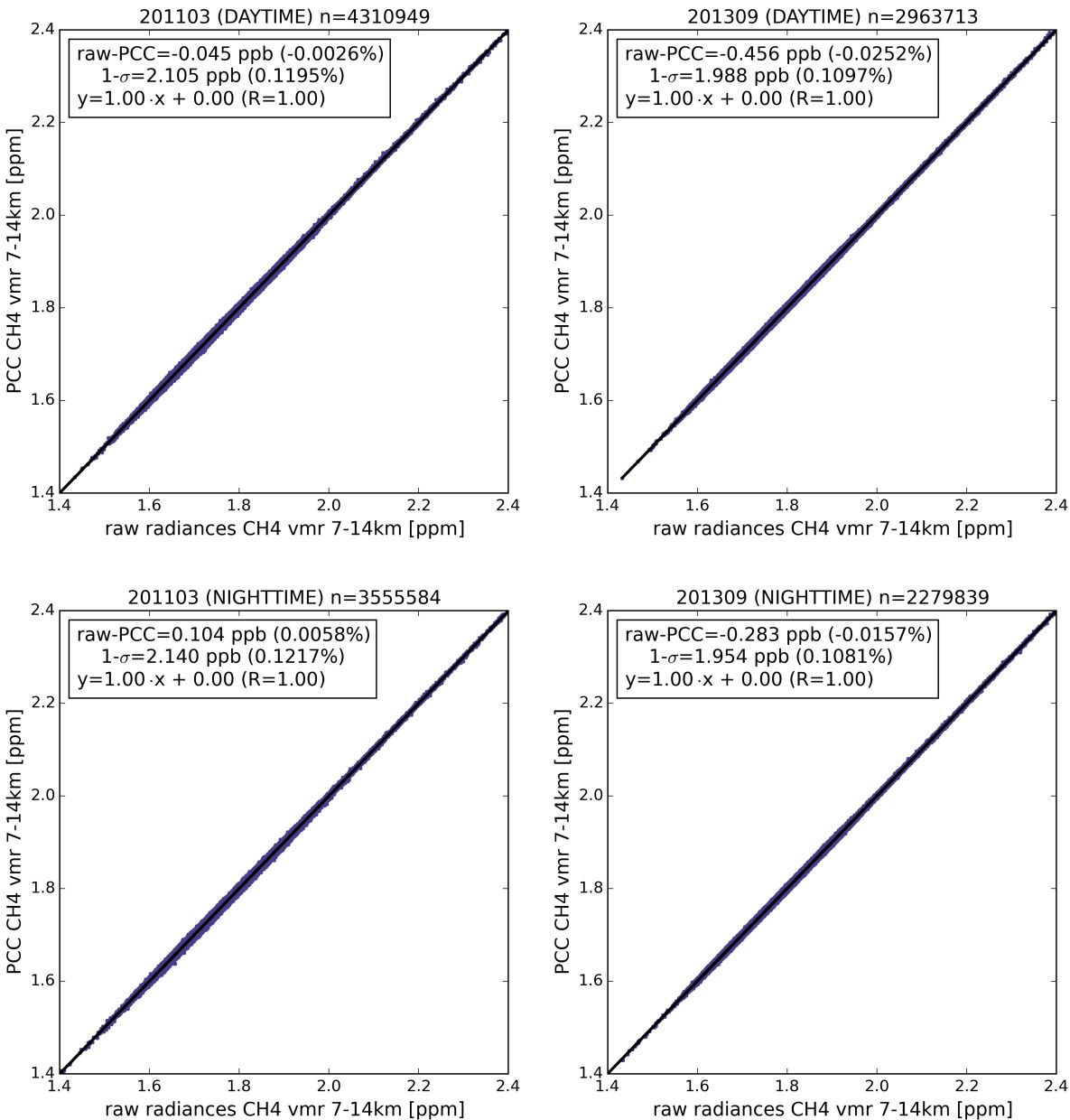

**Figure 4.** Correlation plot retrieved CH$_4$ between 7 and 14 km from the raw radiances (x-axis) and from the PCC-reconstructed radiances (y-axis) for March 2011 [left] and September 2013 [right], for daytime [top] and nighttime [bottom] retrievals between $60°$S and $70°$N. The mean difference and 1-σ of the difference between raw and PCC partial columns is given in ppbv and % in the legend, as well as the slope and intercept from the least-squares fit and correlation coefficient R.

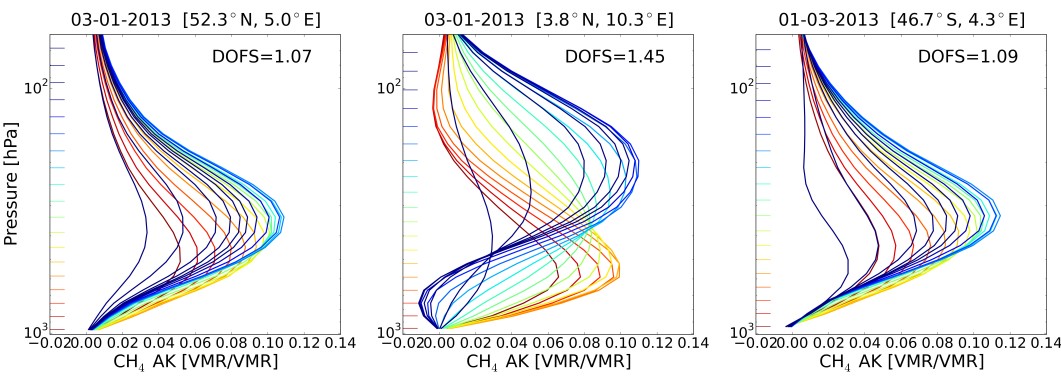

**Figure 5.** CH$_4$ averaging kernels for 3 pixels on the 1$^{st}$ of March 2013 at 3 different locations (52°N, 4°N and 47°S).

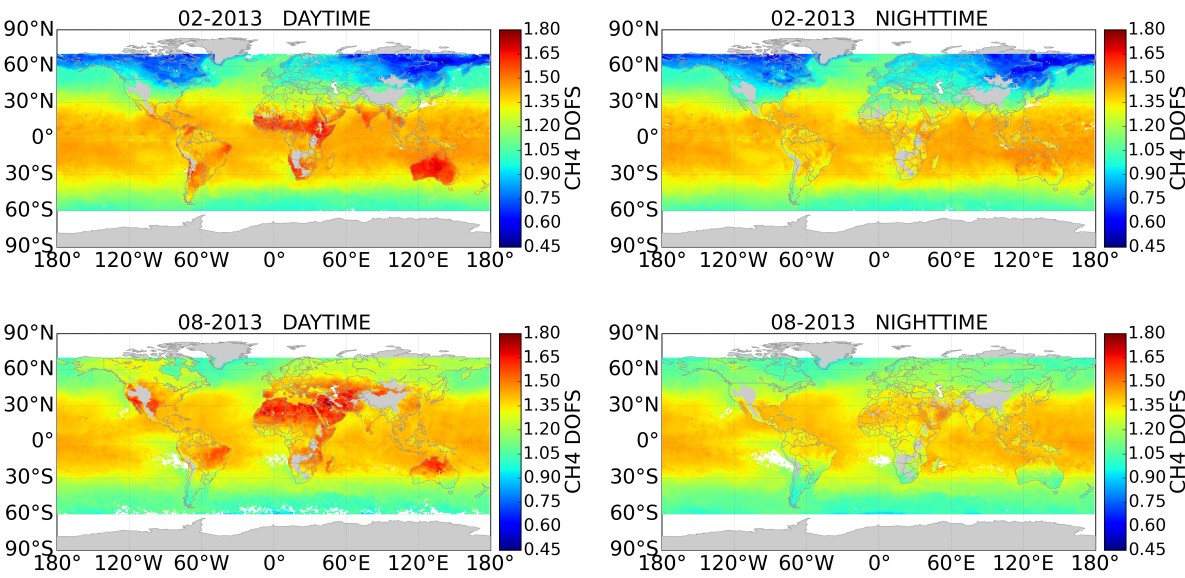

**Figure 6.** Maps of the CH$_4$ Degrees of Freedom for signal (DOFS) calculated from the trace of the CH$_4$ averaging kernel, for February [top] and August [bottom] 2013. DOFS for daytime observations are given on the left, DOFS for nighttime observations on the right.

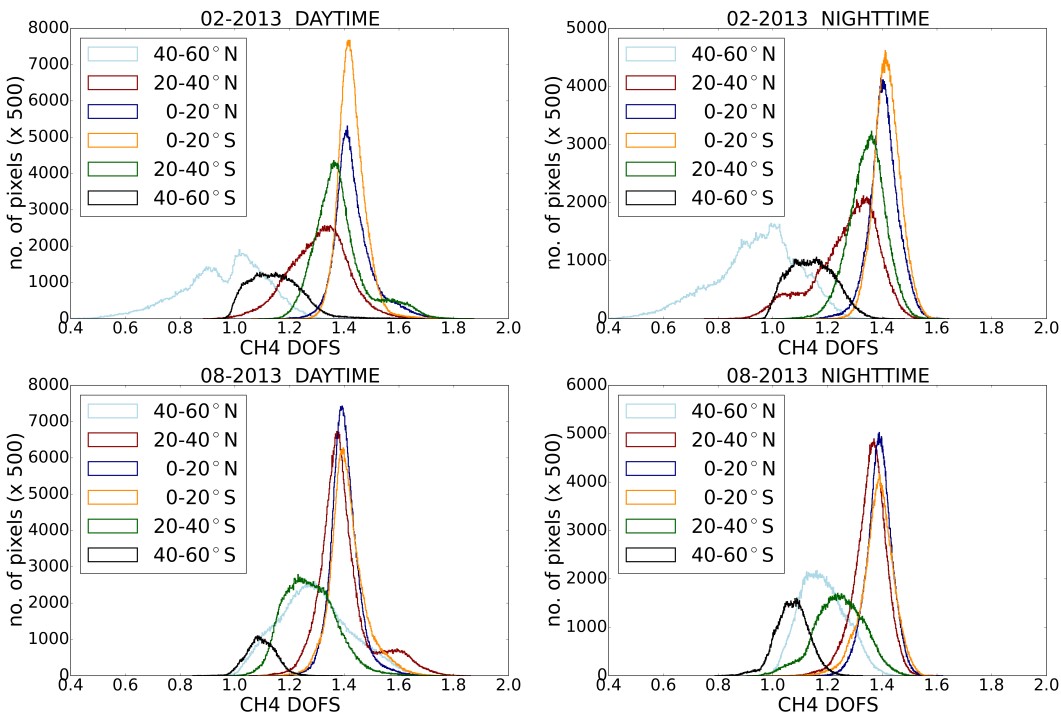

**Figure 7.** CH$_4$ DOFS for different latitudinal bands for daytime [left] and nighttime [right] observations for February [top] and August [bottom] 2013.

| Error Source | Uncertainty Error Source | Uncertainty IASI $CH_4$ |
|---|---|---|
| Temperature profile | - | 1.40% |
| $CH_4$ line intensity | 2% | 1.93% |
| $CH_4$ broadening coefficients | 2% | 1.09% |
| $CH_4$ a priori bias | 2% | 0.06% |
| $H_2O$ line intensity | 2% | 0.02% |
| $H_2O$ broadening coefficients | 2% | 0.03% |
| $N_2O$ line intensity | 2% | 0.05% |
| $N_2O$ broadening coefficients | 2% | 0.03% |
| PCC reconstructed | - | 0.02% |
| Emissivity | 1% | 0.27% |
| Smoothing | a priori variability | 2.45% |
| Measurement noise | $2 \cdot 10^{-8} W/(cm^2\ sr\ cm^{-1})$ | 0.95% |
| Total | | 3.73% |

**Table 2.** The different error sources (column 1) and their uncertainties considered (column 2) for the IASI $CH_4$ uncertainty estimation. The results of the uncertainty estimation of the $CH_4$ 4-17 km partial columns by the perturbation method described in Sect. 4.3 are given in column 3. The uncertainty of the temperature profile on the $CH_4$ 4-17 km partial column is estimated by replacing the IASI L2 temperature profiles with the ECMWF Era-interim re-analysis temperature profiles. To estimate the uncertainty of the PCC reconstructed spectra on the $CH_4$ columns we used the raw spectra and compared the retrieved $CH_4$ partial columns with the PCC reconstructed retrieved $CH_4$ as is done in Sect. 3.3.

| site | LAT | $\Delta$[%] | $\sigma$ [%] | $\epsilon_{sys}$ [%] | $\epsilon_{rand}$ [%] | R | n |
|---|---|---|---|---|---|---|---|
| Eureka | 80°N | -1.45 | 2.05 | 7.63 | 2.65 | 0.67 | 373 |
| Thule | 77°N | 4.04 | 2.02 | 5.28 | 2.58 | 0.73 | 209 |
| Kiruna | 68°N | 0.18 | 2.22 | 3.59 | 2.71 | 0.84 | 437 |
| Jungfraujoch | 47°N | -0.95 | 2.07 | NA | NA | 0.81 | 674 |
| Zugspitze | 47°N | -0.38 | 2.48 | 2.21 | 2.56 | 0.68 | 2020 |
| Toronto | 44°N | -2.31 | 2.97 | 8.26 | 3.21 | 0.52 | 535 |
| Izana | 28°N | -1.19 | 1.76 | 3.29 | 2.46 | 0.37 | 3290 |
| Mauna Loa | 20°N | -0.17 | 1.81 | 4.75 | 2.48 | 0.33 | 592 |
| Maido | 21°S | -0.10 | 2.54 | 3.44 | 2.89 | 0.15 | 478 |
| Wollongong | 34°S | -0.53 | 2.30 | 6.70 | 6.21 | 0.60 | 2230 |

**Table 3.** Statistics of the comparison between the IASI and smoothed NDACC $CH_4$ 4-17 km partial columns for the period 2011-2014. For each location, the latitude coordinates, the mean percentage difference ($\Delta$=(IASI-NDACC)/NDACC) and standard deviation of the difference ($\sigma$), the mean systematic ($\epsilon_{sys}$) and random uncertainty of the differences ($\epsilon_{rand}$), the correlation coefficient (R) and the number of observations (n) are given. NA=not available, for Jungfraujoch the systematic and random uncertainty covariance matrices are not available.

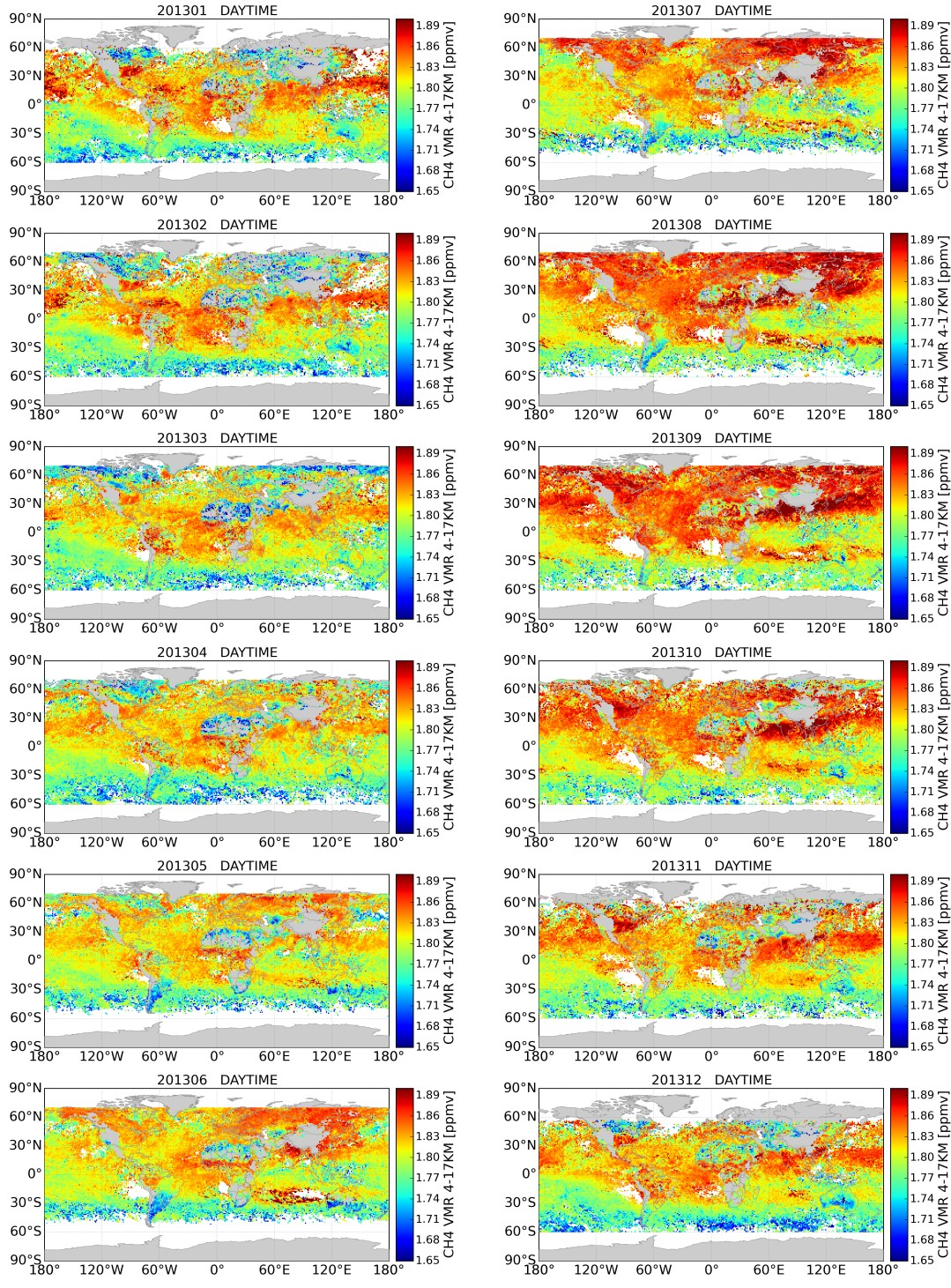

**Figure 8.** Monthly mean global daytime distributions of CH$_4$ partial columns (4-17 km) in 2013. Given is the average over the 2 x 2 circular pixels which are measured simultaneously by IASI. The pixels are binned on a 1° x 1° grid.

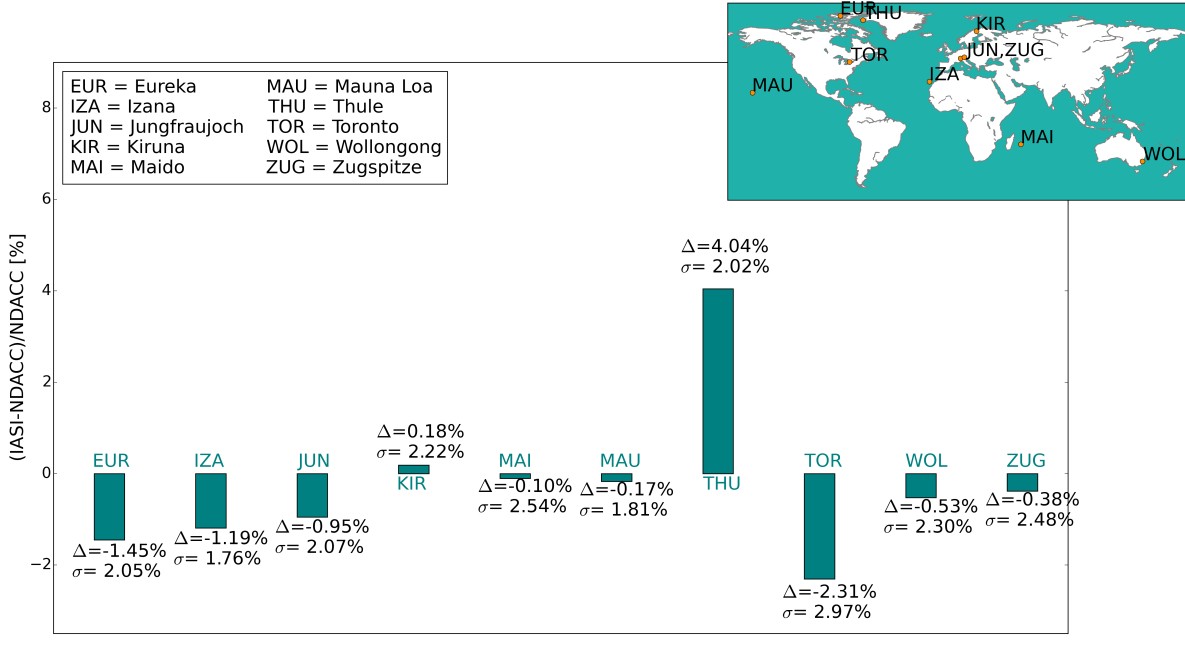

**Figure 9.** Barchart of the results of the IASI-NDACC validation exercise. Given is the relative percentage difference $\Delta$=(IASI-NDACC)/NDACC and standard deviation of the difference ($\sigma$) of partial columns in the 4-17 km altitude range for each of the 10 investigated NDACC sites, visualized in the map on the top right. These results are also summarized in Table 3.

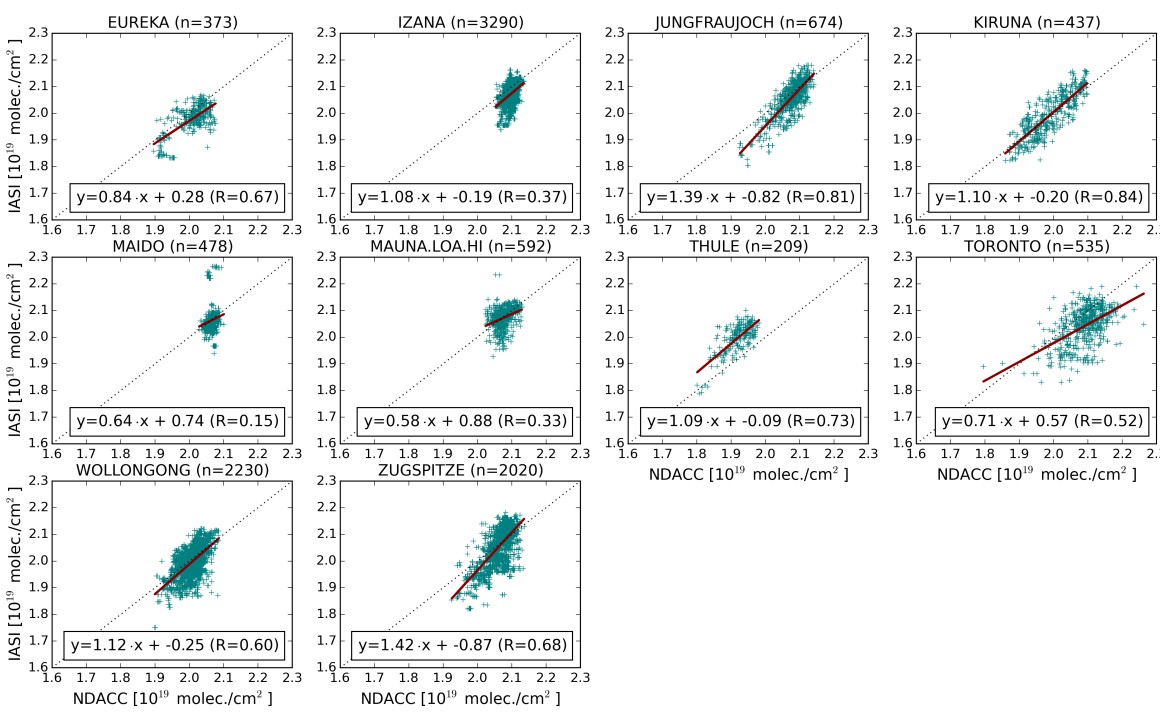

**Figure 10.** Correlation plots of smoothed NDACC and IASI $CH_4$ partial columns (4-17 km) in molec./cm$^2$ for the period 2011-2014. The number of collocations (n) is given for each site in the title. The red lines are the linear regressions between the data points and the dashed black line is the unity slope, shown for comparison. The values of the linear regression and the correlation coefficient (R) are given for each station, the latter is summarized in Table 3.