# Peer review of "Retrieval and validation of MetOp/IASI methane"

_Atmospheric Measurement Techniques, 2017_

## Referee Comment (RC1) · Anonymous Referee #1 · 12 Jul 2017

General comments

Overall the paper is very well written and introduces a new IASI CH4 retrieval. This however is not the first methane data produced from IASI so some discussion on the existing retrievals (Crevoisier 2009; Siddans 2016; Garcia, 2017) and how this work differs or builds on these would be beneficial.

The paper provides a detailed validation assessment against NDACC stations but there is a lack of discussion on what these validation results imply. Is this IASI data useful for ultimately estimating sources and sinks of methane? Or does it lack the necessary precision and accuracy? Some discussion of user requirements here is needed (see e.g. GHG-CCI User Requirements Document).

Validation comparisons against the TCCON network would have also been useful to include in addition to the NDACC data and I'd recommend adding these if that were possible. The paper discusses various next steps for validation (such as against Air-Core measurements) and these are all potentially very valuable. Is there a reason that they have not been performed for this paper? If there was a good reason that these have not been performed then I'd accept that but simply "saving" them for another publication when they'd add a lot of value here is probably not a justifiable reason.

There is now a long timeseries of IASI measurements so for example, looking at the validation against NDACC/TCCON over time would be of value rather than just a scatter plot that loses any temporal information.

Overall I'm happy to accept this paper for publication but I'd like to see the work put more into context of both the existing literature (and pre-existing IASI CH4 datasets) and the scientific usefulness of this data (i.e. estimating sources/sinks). This latter point might require further work on the validation aspect of the paper or at least a discussion on the utility of the data.

Minor comments

Page1 Line 14 – Name methane (CH4) and carbon dioxide (CO2) first time they are mentioned.

P2L2 – Are you referring to the IPCC scenarios? Needs to be clearer.

P4L24 – As methane has a steep gradient above the tropopause, the a priori and covariance can become important. Some more discussion on this aspect would be useful. In particularly, how well does WACCM generally reproduce the atmospheric profile.

P5L5 – Typo "constrained with of an a priori"

P5L11 – Typo "su rface"

P7L8 – Why only 2011-2014 and not the full IASI time period?

Figure 4 – Why only show daytime retrievals?

---

## Referee Comment (RC2) · Anonymous Referee #2 · 21 Jul 2017

SUMMARY

The paper describes the new CH4 product produced by BIRA retrieved from IASI satellite measurements, reconstructed from principal components. The retrieval set-up is described along diagnostics such as averaging kernels and DOFS. The retrievals are compared with NDACC ground-based sites using the standard transformations required to handle the the different vertical sampling.

GENERAL COMMENTS

My main concern is that, having read the paper in some detail, I am still unsure of the quality of the measurements, regarding both the influence of the a priori and the error analysis.

1) The authors state that the CH4 a priori comes from a WACCM climatology but does this a single global/time average, is it zonal/seasonal, or something else? Assuming it has some latitude/seasonal dependence then the question arises: to what extent are the latitude/seasonal cycles depicted in the maps and NDACC comparisons simply reproducing the a priori variations rather than the CH4 retrieval? One way of testing this would be to subtract the a priori from both the data and the NDACC sites and examine the statistics with which *variations* from the climatology are reproduced. This would be a more accurate measure of the added value of the IASI measurements compared to just assuming climatology. Another test, assuming the climatology has no trend, would be to compare time series with the annual cycles removed, but that would probably require more than 3 years of data. While there are plots showing the a priori (smoothing) error contribution to the individual profile levels, there is nothing equivalent for the a priori contribution to the 3-17km partial column which is presented for NDACC comparisons.

2) For the 'reference' paper for any new satellite dataset I feel there should really be a quantitative 'bottom up' error analysis, ie an formal assessment of the magnitudes of the various error terms based on internal tests, which can then be compared by the authors (or other users), to independent data for a 'top down' approach. That is really lacking in this paper. The errors would presumably include contributions from all of the following: instrument noise, errors in retrieved temperature, surface emissivity errors, residual cloud contamination, a priori biases, concentrations of interfering molecules (including HDO) PCC reconstruction error, spectroscopic errors - see specific comments below. These all seem to be handled in the retrieval as a diagonal covariance matrix of fixed size approx 5x noise, and in the NDACC comparisons not assessed at all. An assessment of random error, or precision, could simply be obtained from the SD of the 2x2 pixels or some other small area where it is assumed the CH4 concentration is relatively uniform. It would be useful to have some figures for all these terms, even if only upper limits, for the 3-17km partial column which seems to be the basic product.

[Figure]

And a couple more suggestions, which I leave to the authors' to include or not:

3) Since both day and night are processed separately, although one would not expect the CH4 to change significantly over a diurnal cycle, I would also like to know if the day and night zonal means are self-consistent within the error budgets.

4) A simpler error analysis could be to use the WACCM profiles to convert both NDACC and IASI to total CH4 column amounts.

SPECIFIC COMMENTS:

P1,L1: The product is described as 'global' but results are only shown for 60S-70N.

P1,L4: 'retrieval uncertainty ... less than 4%'. Be a bit more specific about whether this refers to precision (ie random error), or accuracy (total error, including systematic biases) - ideally quote both.

P1,L10: 'absolute differences ... less than 1%'. Again, not clear what this means, 'Absolute' usually means irrespective of +/- sign, and difference could be anything from single-profile match ups to mean bias throughout the whole dataset.

P2,L29: The other IASI CH4 products, currently cited only in the Conclusion, should also be mentioned here at the start to put this work into its proper context.

P3,L4: The final section also contains a significant description of proposed future work.

P3,L6: Pedantically, since there is mention of MetOp-B and MetOp-C, there should be some mention of MetOp-A. And presumably it's not just IASI that will provide a 15 year dataset but all, or at least most, of the other MetOp payload instruments as well.

P3,L14: 'four spectral bands' - I thought there were only three? (breaks at 1210 and 2000cm-1). Also, emphasise that these actually provide a continuous spectrum, without gaps (unlike several other FTIR instruments).

P4,L2: No reference given for Drummond et al.

[Figure]

P4,L8: The EUMETSAT L2 skin temperature is also used as an input (P4,L23).

P4,L11: Is this the dataset commonly referred to as the 'Wicsonsin' surface emissivity data? And what is used over the ocean?

P4,L18: Is there any evidence that the 10% cloud fraction does not contribute a significant error? One might hope that the skin temperature, or other non-CH4 elements of the state vector, will absorb any residual cloud, but that also depends on the tightness of the a priori constraints. I expect the EUMETSAT skin temperature is retrieved with a very small error, so may not allow for much cloud-compensation within the CH4 retrieval. A plot of bias and SD v CH4 a priori, or zonal mean, or NDACC, as a function of cloud percentage would answer this.

P4,L15: What molecule do these 'problem' features belong to? Are they the CH4 Q-branches? And does the forward model include CH4 line-mixing? Also, have the effects of the variation in HDO been considered?

P4,L16: Setting the radiometric noise to infinity (or very large) is the mathematical way to exclude spectral points from the retrieval, not setting the noise to zero. And if these points are excluded from the fit, in what sense is 'no information lost'?

P4,L22: Is the 23-level state vector an arbitrary choice or is it set by the EUMETSAT L2 or WACCM profiles used as a priori data?

P4,L28: Is there a reason for imposing a uniform a priori uncertainty for H2O rather than using the uncertainty associated with the EUMETSAT H2O product that is actually used for the a priori profile? Even if just for scaling the diagonal elements.

P4,L24: I assume this means that the climatology is some sort of average of the WACCM model output - global,annual mean? monthly zonal mean? - while the co-variance represents the model statistical variability about this mean. If the WACCM output is on the same levels as your retrieval grid that's quite straightforward, but if it isn't then there are a few more steps involved.

P4, L29: The characterisation of forward model errors as a simple scaling of the nominal noise diagonal matrix is certainly convenient but requires a little more justification than just the plain statement presented here. The fact that PCC reconstructed radiances agree within the nominal radiometric noise (Fig 3) for a single spectrum is not in itself sufficient to demonstrate that the reconstruction error is negligible: unlike the radiometric noise the reconstruction error is likely to have significant and persistent correlations with the spectrum itself, so unlike the random noise, the impact will generally not be reduced as 1/sqrt(n), where n is the number of spectral points used.

P5,L1: The shape of the averaging kernels presumably depends significantly on the surface temperature contrast, but that information is not given with the figure. It's hardly surprising that, with DOFS∼1, the profile uncertainty is dominated by the a priori error, or 'smoothing error'. A more useful figure would be the error in a quantity which more realistically represents the retrieval information, eg integrated total or partial column amount, and how this compares with the a priori uncertainty. It certainly makes for a more meaningful comparison with other CH4 retrievals which are on different profile levels.

P5,L2: Here the effective sampling range is defined as 2-16 km but elsewhere 4-17 km columns are used.

P5,L31: From Google maps I conclude that these 3 locations are all over the sea - but it would be helpful to state that in the text or the figure caption. If you have only three examples, I don't think it is useful to present both northern and southern mid-latitudes, which one would expect to be similar (particularly near the equinox). It would be more informative to have different land-air temperature contrasts instead, representing the max/min values shown in Fig 6.

P5,L10: 'thermal contrast' needs to be defined.

P5,L20: The correlation plot Fig 4 is used as evidence that the PC reconstruction error is negligible. However, this is a comparison of absolute CH4 values which, as already

demonstrated in Fig 2, are closely constrained by the a priori, so a good correlation may only represent the fact that the measurements have little influence on the a priori. I would be more convinced by a plot of the correlation of the *differences* with respect to the a priori profile.

P6,L20: 'one independent piece of information is retrieved with good sensitivity'. While DOFS $\sim$ 1, it would be more useful to have some idea of, for example, how this translates to a reduction in the a priori uncertainty for a 4-17km column.

P6,L23: What limits the latitude coverage? Here it says 60S-70N but two of the NDACC comparison sites are higher than 70N.

P6,L23: 'binned on a 1x1 deg grid'. So does this mean the plotted points represent not just an average of the four pixels but an average of all the pixels within the 1x1 box? Or is the binning some other process? And what happens if, say one of the 4 pixels is flagged as cloud-contaminated. Is the average then made of the remaining 3 or is this set of 4 pixels excluded?

P8,L24 gives IASI a priori systematic component as 2% of a priori value. Where does this come from? Just the error in WACCM? No systematic component of IASI retrieval uncertainty is considered.

References: not in alphabetical order, some missing publication year, and inconsistently formatted. Patra 2009 listed in references but not cited in text

Fig 1: Rather than just show a generic piece of spectrum it would be helpful if this figure was also used to show the indidividual contributions of different molecules to this spectral region (eg separate panel with same x-axis).

Fig 2: On the left panel it would be helpful to also have the a priori error bars plotted for comparison.

Table 1: This lists a priori information as WACCM, but that is only for certain elements of the state vector. 'IASI L2' should include the word 'EUMETSAT' for consistency with

the text. There should also be something about the a priori covariance information.

Table 2: I'm surprised at the spread in systematic errors in Table 2 for the various NDACC comparisons. Assuming this represents a combination of the NDACC systematic error budget and the 2% systematic error assumed for the retrieval a priori, this variation must mostly come from the NDACC data. Yet Sepulveda et al (2014) quotes a figure of 2.5% which is largely spectroscopic uncertainty (and therefore common to all sites). And the fact that these systematic errors are all much larger than the biases suggests something wrong.

TYPOGRAPHICAL ERRORS/SUGGESTIONS

P1,L3: 'usefulness'

P2, L2: 'greenhouse gas-intensive' - the hyphen here seems to change the natural coupling of words and might be better removed.

P2, L3 (and elsewhere): 'ppb' - suggest 'ppbv', to distinguish from 'ppbm'.

P2,L5 (and elsewhere): 'Bulletin , 2016' - remove space between reference name and comma

P2,L31: As an acronym, I think 'lidar' now has the same status as 'radar' and need not be capitalised or expanded (at least not for the benefit of AMT readers)

P3,L6: no comma required after 'MetOp'.

P3,L7: 'successive series' - a series is, by definition, successive.

P3,L15: 'prediction', singular, seems more usual in this context.

P3,L19: 'In the longer term...'

P3,L19: 'programme', unless you choose American English.

P3,L20: 'observations' (plural).

P3,L3: 'a modular software'. While 'the software' is acceptable, 'a software' doesn't sound right. I suggest just '.. is modular software ..'.

P4,L3: 'on-board' inconsistent hyphenation cf P3,L6 & L8.

P4,L8: 'range' mentioned twice in same line

P4,L12: 'and the residual'

P3,L14: 'certain spectral ranges ... spectral band'. Previously the 1210-1290 was described as the 'spectral range' but it seems that now becomes the 'spectral band' and 'spectral range' now refers to the problematic spectral features.

P4,L22: 'Table 1'

P4,L28: 'its a priori'. Firstly, unclear what 'its' refers to and, secondly, 'a priori' suggests a retrieved quantity. I suggest just replacing 'its a priori' by 'their' or 'with'.

P4,L32: 'vapor' here, but 'vapour' elsewhere, eg P4,L8.

P5,L3: 'diplays'

P4,L5: 'with of an'

P4,L11: 'su rface'

P5,L17: 'Tb/year' should be reduced to 'Tb' given the rest of the sentence.

P5,L19 & L20: 'm2 sr m-1' should be 'cm2 sr cm-1'

P5,L24: 'negligibles'

P6,L30: 'van' - should be 'Van' since starting a new sentence.

P7,L17: 'less' - should be 'fewer'

P8,L5: 'alitude'

P8,L30: 'slighlty'

P12,L32: 'Forc- ing'

Fig 1: insert space: 'sr cm'
* * *

---

## Author Response (AR1)

We would like to thank the referees for their time and useful comments to help improve this manuscript. Hereby our replies to the different points which were addressed by referee # and #2. The referee comments are given in italic blue, our replies in black. The page and line numbers mentioned refer to the non-revised AMTD manuscript. A revised version of the manuscript highlighting the changes made (with latexdiff) is attached at the end.

In the course of the review, we stumbled on an error in one of the calculations in the validation scripts. This error had an impact on the validation results. Therefore chapter 5 has been adjusted. The figures and table have been updated as well as the text. We sincerely apologize for this.
The changes made are listed at the end after addressing the different referee comments.

**REPLY COMMENTS REFEREE # 1**

**GENERAL COMMENTS**
*Overall the paper is very well written and introduces a new IASI CH4 retrieval. This however is not the first methane data produced from IASI so some discussion on the existing retrievals (Crevoisier 2009; Siddans 2016; Garcia, 2017) and how this work differs or builds on these would be beneficial.*

We added the following description of the other existing products in the Introduction on P2, L33:
As mentioned in the previous paragraph, in addition to the IASI CH4 product presented here, other IASI CH4 products exist. Crevoisier et al. (2009) uses a non-linear inference scheme based on neural networks, to derive a mid-tropospheric CH4 column with peak sensitivity at about 230 hPa (~11 km), half the peak sensitivity at 100 and 500 hPa (~6 and 16 km), and no sensitivity to the surface. This dataset was previously only available for the tropical region between 30°S and 30°N but got extended to higher latitudes and is available through the Climate Change Initiative-Greenhouse Gas (CCI-GHG) project. The retrieval schemes of Siddans et al. (2016) and García et al. (2017) are based on the optimal estimation method (OEM), like the BIRA-IASB product. Different constraint matrices are used by the 2 products. García et al. (2017) et al. use a Tikhonov-Philips slope constraint with strong regularisation (almost equivalent to a scaling retrieval). Siddans et al. (2016) use an a priori covariance matrix which describes the presumed errors in the a priori estimate of CH4. The IASI CH4 product presented in this paper follows a similar approach as Siddans et al. (2016).

*The paper provides a detailed validation assessment against NDACC stations but there is a lack of discussion on what these validation results imply. Is this IASI data useful for ultimately estimating sources and sinks of methane? Or does it lack the necessary precision and accuracy? Some discussion of user requirements here is needed (see e.g. GHG-CCI User Requirements Document).*
The GHG-CCI User Requirements Document outlines the random ("precision") and systematic retrieval error requirements for regional CH4 source/sink determination for dry-air column CH4 measurements over land. They stipulate a threshold random error requirement <34 ppb (~1.8%) for a single observation and a <10 ppb (~0.5%) systematic error (after empirical bias correction that does not use the verification data). However, we found the CMUG-CCI (Climate Modelling User Group - Climate Change Initiative) Requirements Baseline Document (RBD) more useful since it gives estimates of the observational requirements for tropospheric CH4 columns and profiles.

Page 29 of the CMUG-CCI RBD tabulates the GHG observational requirements for regional CH4 source/sink determination. A tropospheric CH4 column observation at a horizontal resolution of 50 km requires a precision of 1% and an accuracy of 2% (and a 6h observing cycle). These requirements are very demanding and are not yet met.

We added the following paragraph to the Conclusion, P10, L3:

Sect. 1 highlighted the need to improve our current understanding of the global budget of CH4. The Requirements Baseline Document (RBD) of the CCI-Climate Modelling User Group (CMUG) stipulates the observational requirements for regional source/sink determination of CH4. The RBD states that a CH4 profile/tropospheric CH4 column observation at a horizontal resolution of 50 km requires a precision of 1% and an accuracy of 2% for a 6h observing cycle (CMUG-RBD, 2015). To reach these demanding requirements improvements in the precision of the IASI 4-17 km partial column are needed. The error analysis in Section 4.3 gives a random uncertainty of 2.98% with the largest contribution coming from the smoothing error and uncertainty of the temperature profile. The simultaneous retrieval of the temperature profile could improve the IASI precision and will be investigated in the near future. The systematic uncertainty estimate in Section 4.3 of 2.23% could be improved by reducing spectroscopic uncertainties. Continuous efforts will be made on improving the IASI CH4 retrievals as to these issues and enhancing their precision.

*Validation comparisons against the TCCON network would have also been useful to include in addition to the NDACC data and I'd recommend adding these if that were possible. The paper* discusses various next steps for validation (such as against AirCore measurements) and these are all potentially very valuable. Is there a reason that they have not been performed for this paper? If there *was a good reason that these have not been performed then I'd accept that but simply "saving" them for another publication when they'd add a lot of value here is probably not a justifiable reason. There* is now a long timeseries of IASI measurements so for example, looking at the validation against NDACC/TCCON over time would be of value rather than just a scatter plot that loses any temporal information.

The TCCON network is the reference network for the validation of GHG satellite data. Specifically for shortwave infrared (SWIR) satellite data, which have a similar vertical sensitivity as the TCCON measurements.

The vertical sensitivity of the TCCON measurements is different to the vertical sensitivity of thermal infrared (TIR) sounders like IASI. TCCON is sensitive in the troposphere with a good sensitivity at the surface while TIR sounders are sensitive to the upper troposphere-lower stratosphere and have limited sensitivity at the surface. Smoothing the higher resolved TCCON data with the IASI averaging kernel (AK) is not possible since TCCON does not deliver a retrieved TCCON profile, only the dry-air column. Therefore we would be comparing 2 measurements with a different sensitivity. I therefore am in no favour at all of validating the IASI CH4 data with the TCCON network.

Regarding the AirCore data, we actually don't have sufficient AirCore data. We only got in possession of about 2 dozen AirCore profiles at Sodänkyla shortly after the first draft of this paper and have not done the comparison yet. We are in contact with another group who might deliver AirCore campaign data from (a) different site(s) but we have nothing concrete yet. It is surely not our intention to 'save' this for another publication.

*Overall I'm happy to accept this paper for publication but I'd like to see the work put more into context* of both the existing literature (and pre-existing IASI CH4 datasets) and the scientific usefulness of this data (i.e. estimating sources/sinks). This latter point might require further work on the validation aspect of the paper or at least a discussion on the utility of the data.

We've added an additional section (Section 4.3) to the manuscript giving a more detailed error analysis of the IASI CH4 columns to also help determine the scientific usefulness of this data. In addition we extended the discussion on other existing IASI CH4 retrievals.

We hope hereby the referee and the reader have the necessary information and we very much believe that the referee's suggestions have improved the manuscript.

MINOR COMMENTS:

P1, L14: Name methane (CH4) and carbon dioxide (CO2) first time they are mentioned.
This has been implemented. We changed the following:
..increased emissions of greenhouse gases like CO2 and CH4 since the pre-industrial era..
→
..increased emissions of greenhouse gases like carbon dioxide (CO2) and methane (CH4) since the pre-industrial era..

P2, L2: Are you referring to the IPCC scenarios? Needs to be clearer.
Indeed, the Representative Concentration Pathways, as introduced by the IPCC. We adjusted the sentence as follows:
Since 2014, atmospheric CH4 concentrations are rising faster than at any time in the past two decades and are now approaching the most greenhouse gas-intensive scenarios (Saunois et al., 2016).
→
Since 2014, atmospheric CH4 concentrations are rising faster than at any time in the past two decades and its concentration is now approaching the most greenhouse gas intensive Representative Concentration Pathway (RCP) trajectories (Saunois et al., 2016), the scenario pathways which were introduced by the Intergovernmental Panel on Climate Change (IPCC) in its fifth Assessment Report (AR5) in 2014.

P4,L24: As methane has a steep gradient above the tropopause, the a priori and covariance can become important. Some more discussion on this aspect would be useful. In particularly, how well does WACCM generally reproduce the atmospheric profile.
The a priori profiles xa and covariance matrices Sa for CH4 and N2O are based on a climatology from the WACCM model. A single CH4 a priori profile, representative for a mid-latitude CH4 profile, is used for all latitudes and seasons. The CH4 covariance matrix represents the highest variability at the surface and in the upper troposphere-lower stratosphere (UTLS). The variability in the UTLS is representative for the variability of the CH4 gradient at the tropopause which is different at different latitudes.
The other referee also asked for additional information, so we added the following text in the manuscript on P4, L24:

The a priori profiles xa and covariance matrices Sa for CH4 and N2O are based on a climatology from the WACCM model. A single CH4 a priori profile, representative for a mid-latitude CH4 profile, is used for all latitudes and seasons. Therefore the atmospheric CH4 variations observed are a results of the variability of atmospheric CH4 rather than the a priori information. The CH4 covariance matrix represents the highest variability at the surface and in the upper troposphere-lower stratosphere (UTLS). The variability in the UTLS is representative for the variability of the CH4 gradient at the tropopause which is different at different latitudes.

*P5, L5: Typo "constrained with of an a priori"*
P5 L5: Thank you, this has been corrected.

*P5, L11 : Typo "su rface"*
P5 L11: Has been corrected.

P7, L8: Why only 2011-2014 and not the full IASI time period?

Mostly due to processing limitations. IASI measures about 1 million spectra a day. So far we processed the data necessary for the projects we're involved in.

Figure 4: Why only show daytime retrievals?
Since the results are similar. We added the comparison for the nighttime retrievals in Figure 4 and adjusted the text and caption accordingly.

P5, L22:
..for March 2011 and September 2013 for daytime and retrievals..
→
..for March 2011 and September 2013 for daytime and nighttime retrievals..

Caption Figure 4:
..the PCC-reconstructed radiances (y-axis) for March 2011 [left] and September 2013 [right], for daytime retrievals between 60° and 70°N.
→
..the PCC-reconstructed radiances (y-axis) for March 2011 [left] and September 2013 [right], for daytime [top] and nighttime [bottom] retrievals between 60° and 70°N.

[Figure]

**Figure 4.** Correlation plot retrieved CH4 between 7 and 14 km from the raw radiances (x-axis) and from the PCC-reconstructed radiances (y-axis) for March 2011 [left] and September 2013 [right], for daytime [top] and nighttime [bottom] retrievals between 60°S and 70°N. The mean difference and 1-σ of the difference between raw and PCC partial columns is given in ppbv and % in the legend, as well as the slope and intercept from the least-squares fit and correlation coefficient R.

References

GHG-CCI User Requirements Document : online available at http://www.esa-ghg-cci.org/?q=webfm_send/344

CMUG-CCI Requirements baseline document : online available at http://ensembles-eu.metoffice.com/cmug/CMUG_PHASE_2_D1.1_Requirements_v0.6.pdf

**REPLY COMMENTS REFEREE # 2**

**GENERAL COMMENTS**

My main concern is that, having read the paper in some detail, I am still unsure of the quality of the measurements, regarding both the influence of the a priori and the error analysis.

We will hereby address these point in order to give clarity to the referee on the quality of our product by outlining a more detailed error analysis and by showing that the atmospheric CH4 variations observed are a result of the variability of atmospheric CH4 rather than the a priori information.

1) The authors state that the CH4 a priori comes from a WACCM climatology but does this a single global/time average, is it zonal/seasonal, or something else? Assuming it has some latitude/seasonal dependence then the question arises: to what extent are the latitude/seasonal cycles depicted in the maps and NDACC comparisons simply reproducing the a priori variations rather than the CH4 retrieval? One way of testing this would be to subtract the a priori from both the data and the NDACC sites and examine the statistics with which *variations* from the climatology are reproduced.

This would be a more accurate measure of the added value of the IASI measurements compared to just assuming climatology. Another test, assuming the climatology has no trend, would be to compare time series with the annual cycles removed, but that would probably require more than 3 years of data. While there are plots showing the a priori (smoothing) error contribution to the individual profile levels, there is nothing equivalent for the a priori contribution to the 3-17km partial column which is presented for NDACC comparisons.

This was indeed not clearly mentioned in the manuscript.
The a priori profile is a single CH4 profile used for all latitudes and seasons. It therefore shows no latitudinal/seasonal dependence. The choice for a single a priori profile is, as the referee mentions, to **not** introduce any variability in the final retrieved product from the a priori variability. Below we show 4 figures. On the left hand side 2 figures of retrieved IASI CH4 for January 2013 and July 2013 (as in Fig. 8) and on the right hand side the CH4 a priori is given, which is constant at all latitudes and for the different seasons. As you can see, we use one single CH4 a priori profile.
In Sect. 3.2 we added the following text to make sure there is no unclarity regarding the a priori profile used :
A single global CH4 xa profile is used for all the retrievals, representative of a mid-latitude CH4 profile. Therefore the atmospheric CH4 variations observed are a results of the variability of atmospheric CH4 rather than the a priori information.

[Figure]

Review-Figure 1 : [left] Monthly mean global daytime distribution of CH4 partial columns (4-17 km) in 2013, as given in Figure 8 of the manuscript. [right] A priori CH4 partial columns (4-17 km) for the same 2 months.

*2) For the 'reference' paper for any new satellite dataset I feel there should really be a quantitative 'bottom up' error analysis, ie an formal assessment of the magnitudes of the various error terms based* on internal tests, which can then be compared by the authors (or other users), to independent data for *a 'top down' approach. That is really lacking in this paper. The errors would presumably include* contributions from all of the following: instrument noise, errors in retrieved temperature, surface emissivity errors, residual cloud contamination, a priori biases, concentrations of interfering molecules (including HDO) PCC reconstruction error, spectroscopic errors - see specific comments below. These all seem to be handled in the retrieval as a diagonal covariance matrix of fixed size approx 5x noise, and in the NDACC comparisons not assessed at all. An assessment of random error, or precision, could simply be obtained from the SD of the 2x2 pixels or some other small area where it is assumed the CH4 concentration is relatively uniform. It would be useful to have some figures for all these terms, even if only upper limits, for the 3-17km partial column which seems to be the basic product.

We added a section Error Analysis in which we calculate the error budget of the IASI CH4 4-17 km partial column.
Here we include the values of the smoothing and measurement uncertainties as estimated following the formalism by Rodgers (2000) and as was discussed in Section 3.2. Additionally we estimate the uncertainty on the CH4 4-17km partial column by additional error sources such as the temperature profile, the emissivity and spectroscopic parameters by a perturbation method. This analysis gives an

estimate of the random and systematic uncertainty of the IASI CH4 4-17km partial column product.

The parameters considered in the state vector; the interfering species H2O and N2O, and the skin temperature have no significant contribution to the uncertainty of the CH4 product. This can be seen from the contribution of these species to the CH4 averaging kernel. In the figure below the CH4 averaging kernel (AK) is given (left), and the contribution of the other parameters in the state vector on the CH4 AK, the H2O profile (2nd figure), the N2O profile (3rd figure) and the skin temperature Ts (figure on the right).

[Figure]

Review-Figure 2: [left] IASI CH4 averaging kernel (AK), as is given in Figure 2 of the manuscript. The contribution of the additional parameters in the state vector to the CH4 AK; H2O [2nd figure from the left], N2O [3rd figure from the left], and the skin temperature (Ts) [right].

Nevertheless, we considered the uncertainty of the spectroscopic parameters of the 2 dominant interfering species N2O and H2O + isotopes and found an insignificant contribution to the CH4 retrieved values. The PCC uncertainty is estimated by comparing the difference in retrieved CH4 from the PCC radiances and the raw radiances.

The following section (Section 4.3) discussing the error sources and their contribution was added to the manuscript:

[revised manuscript text omitted]

*And a couple more suggestions, which I leave to the authors' to include or not:*

3) Since both day and night are processed separately, although one would not expect the CH4 to change significantly over a diurnal cycle, I would also like to know if the day and night zonal means are self-consistent within the error budgets.

Indeed, one does not expect for CH4 to change significantly over a diurnal cycle. The differences between day and nighttime retrieved CH4 would mostly be due to differences in sensitivity due to different thermal contrast conditions. In the figure below (Figure 3) the daytime and nighttime zonal means are given for 2 months, February and August 2013, as well as their relative differences. The mean relative difference between daytime and nighttime zonal means is -0.29% and 0.05% for February and August 2013 respectively. These values are well below the uncertainties estimated from the error analysis of Section 4.3 (2.63% considering only the smoothing and measurement uncertainty and 3.73% considering the additional error sources as temperature profile, emissivity and spectroscopy). We decided not to include this in the manuscript, but hope the referee finds these results satisfactory.

[Figure]

Review-Figure 3: [left] Zonal mean CH4 4-17 km partial columns in vmr for February and August 2013 for daytime (yellow) and nighttime (purple) measurements. [right] Relative differences of daytime and nighttime zonal mean measurements (solid line) with the 1-sigma standard deviation given by the colored area. The mean difference and standard deviation of the difference are given in the title.

4) A simpler error analysis could be to use the WACCM profiles to convert both NDACC and IASI to total CH4 column amounts.

Thank you for the suggestion but we decided that the error analysis based on the perturbation method outlined in Section 4.3 is a more detailed and thorough estimate of the CH4 error budget.

*P1,L1: The product is described as 'global' but results are only shown for 60S-70N.*

Indeed. The term 'global' has been removed.

A new global IASI methane product developed at ..

→

A new IASI methane product developed at ..

*P1,L4: 'retrieval uncertainty ... less than 4%'. Be a bit more specific about whether this refers to precision (ie random error), or accuracy (total error, including systematic biases) - ideally quote both.*

With the Error Analysis added (Section 4.3) we changed this to:

The retrieval uncertainty of the CH4 profiles is less than 4% below 100 hPa (~16 km).

→

An detailed error analysis was performed. The total uncertainty is estimated to be 3.73% for a CH4 partial column between 4-17 km.

*P1,L10: 'absolute differences ... less than 1%'. Again, not clear what this means, 'Absolute' usually means irrespective of +/- sign, and difference could be anything from single-profile match ups to mean bias throughout the whole dataset.*

We changed this and also added the term 'Relative mean differences'. Changes were also made with respect to the revised validation results.

P1, L9:

Mean differences between IASI and FTIR CH4 range between -1.93 and 4.40% and are within the systematic uncertainty.

For 7 out of the 10 stations absolute differences are less than 1%.

→

Relative mean differences between IASI and FTIR CH4 range between -2.31 and 4.04% and are within the systematic uncertainty. For 6 out of the 10 stations the relative mean differences are smaller than ±1%.

*P2,L29: The other IASI CH4 products, currently cited only in the Conclusion, should also be mentioned here at the start to put this work into its proper context.*

This has been added :

..METOP-B in September 2012 (Razavi et al., 2009; Crevoisier et al., 2009; Xiong et al., 2013).

→

..METOP-B in September 2012 (Razavi et al., 2009; Crevoisier et al., 2009; Xiong et al., 2013; Siddans et al., 2016; Garcia et al., 2017).

*P3,L4: The final section also contains a significant description of proposed future work.*

Indeed, thank you, we added this.

The final section summarizes the main results of this work.

→

The final section summarizes the main results of this work and discusses future work.

*P3,L6: Pedantically, since there is mention of MetOp-B and MetOp-C, there should be some mention of MetOp-A. And presumably it's not just IASI that will provide a 15 year dataset but all, or at least most, of the other MetOp payload instruments as well.*

We changed MetOp to MetOp-A.

The Infrared Atmospheric Sounding Interferometer (IASI) onboard MetOp is a thermal..

$\rightarrow$

The Infrared Atmospheric Sounding Interferometer (IASI) onboard MetOp-A is a thermal..

*P3,L14: 'four spectral bands'* - I thought there were only three? (breaks at 1210 and 2000cm-1). Also, emphasise that these actually provide a continuous spectrum, without gaps (unlike several other FTIR instruments).

Indeed, thank you for pointing that out. We changed the following:

IASI has four spectral bands in the spectral range from 645 to 2760 cm-1 ..

$\rightarrow$

IASI has three spectral bands in the spectral range from 645 to 2760 cm-1 (3.62 to 15.5 µm), provided as a continuous spectrum with an apodized spectral resolution of 0.5 cm-1 and spectral sampling of 0.25 cm-1.

P4,L2: No reference given for Drummond et al.

Thank you. We added the reference of Drummond et al.:

Drummond, R., Vandaele, A.-C., Daerden, F., Fussen, D., Mahieux, A., Neary, L., Neefs, E., Robert, S., Willame, Y., and Wilquet, V.: Studying methane and other trace species in the Mars atmosphere using a SOIR instrument, Planetary and Space Science, 59, 292–298, http://www.sciencedirect.com/science/article/pii/S0032063310001479, 2011

P4,L8: The EUMETSAT L2 skin temperature is also used as an input (P4,L23).

We changed this.

EUMETSAT IASI L2 temperature and water vapour profiles are used as input for the radiative transfer calculations.

$\rightarrow$

EUMETSAT IASI L2 skin temperature (Tskin), temperature and water vapour profiles are used as input for the radiative transfer calculations.

P4,L11: Is this the dataset commo*nly referred to as the 'Wicsonsin' surface emissivity data? And what* is used over the ocean?

I'm not familiar how this dataset is commonly referred to, but I don't think it's the same one. The Zhou emissivity/climatology database generated from IASI measurements (0.5°x0.5°) we received from Dan Zhou includes emissivity values over land and over water.

*P4,L15: What molecule do these 'problem' features belong to? Are they the CH4 Q-*branches? And does the forward model include CH4 line-mixing? Also, have the effects of the variation in HDO been considered?

We have not been able to identify that yet. It could be a combination of line mixing or uncertainties in the HDO spectroscopy.

Our forward model does not include CH4 line mixing. Razavi. et al. (2009) investigated the impact of line mixing in the nu4 spectral band of CH4. They showed that the most influential impact was around the methane Q branch around 1306 cm-1 and they therefore excluded the Q branch from the retrieval window. We followed their reasoning.

These features are not part of the Q branch, this spectral region is the CH4 nu4 P branch.

The isotopologues of the different molecules included are considered in our forward model, so we do include HDO.

*P4,L16: Setting the radiometric noise to infinity (or very large) is the mathematical way to exclude spectral points from the retrieval, not setting the noise to zero. And if these points are excluded from the fit, in what sense is 'no information lost'?*

Sorry, indeed, we mean the signal to noise is set to 0, thank you for pointing this out. This has been corrected. If you would take several spectral windows, to exclude these spectral points, instead of the whole spectral region we are taking now (1210-1290 cm-1), your information content will reduced. By masking these spectral points we don't loose information content.

We changed the following:

These spectral ranges are masked in the retrieval set-up, i.e. the radiometric noise is set to zero at these spectral points..

→

These spectral ranges are masked in the retrieval set-up, i.e. the signal-to-noise is set to zero at these spectral points..

*P4,L18: Is there any evidence that the 10% cloud fraction does not contribute a significant error? One might hope that the skin temperature, or other non-CH4 elements of the state vector, will absorb any residual cloud, but that also depends on the tightness of the a priori constraints. I expect the EUMETSAT skin temperature is retrieved with a very small error, so may not allow for much cloud-compensation within the CH4 retrieval. A plot of bias and SD v CH4 a priori, or zonal mean, or NDACC, as a function of cloud percentage would answer this.*

What we actually see is that taking a fractional cloud cover < 10% based on the EUMETSAT IASI L2 fractional cloud fraction corresponds to a 0% fractional cloud coverage. We therefore can't generate a figure of the retrieved CH4 as a function of cloud percentage.

We'd like to point out that many other trace gas retrieval products of IASI use a less constrained filtering of cloud contaminated spectra. F.e. for the IASI $NH_3$ (Van Damme et al. 2014) and $HNO_3$ (Ronsmans et al., 2016) products spectra with a fractional cloud cover <25 % are processed.

*P4,L22: Is the 23-level state vector an arbitrary choice or is it set by the EUMETSAT L2 or WACCM profiles used as a priori data?*

The 23-level state vector is an arbitrary choice.

*P4,L28: Is there a reason for imposing a uniform a priori uncertainty for H2O rather than using the uncertainty associated with the EUMETSAT H2O product that is actually used for the a priori profile? Even if just for scaling the diagonal elements.*

The EUMETSAT $H_2O$ product has an accuracy of 10% which we set here as the standard deviation of the IASI L2 $H_2O$ a priori. Initial tests with different $H_2O$ uncertainties/covariance matrices, gave us the most optimal retrieval results for a uniform covariance matrix with a 10% standard deviation and correlation length of 6 km.

*P4,L24: I assume this means that the climatology is some sort of average of the WACCM model output - global,annual mean? monthly zonal mean? - while the covariance represents the model statistical variability about this mean. If the WACCM output is on the same levels as your retrieval grid that's quite straightforward, but if it isn't then there a*re a few more steps involved.

A single global CH4 xa profile is used for all the retrievals, representative of a mid-latitude CH4 profile. The other referee also asked for additional information, so we added the following text in the manuscript on P4, L24:

The a priori profiles xa and covariance matrices Sa for CH4 and N2O are based on a climatology from the WACCM model. A single CH4 a priori profile, representative for a mid-latitude CH4 profile, is used for all latitudes and seasons. Therefore the atmospheric CH4 variations observed are a results of

the variability of atmospheric CH4 rather than the a priori information. The CH4 covariance matrix represents the highest variability at the surface and in the upper troposphere-lower stratosphere (UTLS). The variability in the UTLS is representative for the variability of the CH4 gradient at the tropopause which is different at different latitudes.

P4, L29: The characterisation of forward model errors as a simple scaling of the nominal noise diagonal matrix is certainly convenient but requires a little more justification than just the plain statement presented here. The fact that PCC reconstructed radiances agree within the nominal radiometric noise (Fig 3) for a single spectrum is not in itself sufficient to demonstrate that the reconstruction error is negligible: unlike the radiometric noise the reconstruction error is likely to have significant and persistent correlations with the spectrum itself, so unlike the random noise, the impact will generally not be reduced as 1/sqrt(n), where n is the number of spectral points used.
As discussed in general comment #2, a section with a more detailed error analysis of the IASI CH4 4-17 km partial column has now been added (Section 4.3).
Indeed, the reconstruction process is likely to introduce spectral correlations. However, comparing the retrieved partial columns of CH4 of PCC spectra with raw radiances in the perturbation theory and as shown in Section 3.3 (Figure 4), we found this error source negligible in comparison to other error sources such as spectroscopy, the temperature profile, emissivity..

P5,L1: The shape of the averaging kernels presumably depends significantly on the surface temperature contrast, but that information is not given with the figure. It's hardly surprising that, with DOFS~1, the profile uncertainty is dominated by the a priori error, or 'smoothing error'. A more useful figure would be the error in a quantity which more realistically represents the retrieval information, eg integrated total or partial column amount, and how this compares with the a priori uncertainty. It certainly makes for a more meaningful comparison with other CH4 retrievals which are on different profile levels.
The uncertainties are now added for the 4-17 km partial columns in Section 4.3. The uncertainty values of the different error sources are listed in Table 2 to give a more meaningful comparison with other CH4 retrieval products.

P5,L2: Here the effective sampling range is defined as 2-16 km but elsewhere 4-17 km columns are used.
The wording of this sentence make it seem like a generalized statement for the whole IASI CH4 dataset indeed. We changed the sentence as follows:
It shows that the sensitivity of the IASI CH4 product lies in the 800-100 hPa (~2-16 km) range.
→
It shows that the sensitivity of the retrieved IASI CH4 profile lies in the 800-100 hPa (~2-16 km) range.

P5,L20: The correlation plot Fig 4 is used as evidence that the PCC reconstruction error is negligible. However, this is a comparison of absolute CH4 values which, as already demonstrated in Fig 2, are closely constrained by the a priori, so a good correlation may only represent the fact that the measurements have little influence on the a priori. I would be more convinced by a plot of the correlation of the *differences* with respect to the a priori profile.
We're not 100% sure we understand the referee correctly here. Under the assumption we use a variable a priori CH4 profile (over location and time), indeed the correlation may represent the variability in the a priori if we're not sensitivity to the measurement. However this is not the case, we use 1 single a priori profile. Looking at the correlation of the difference wrt the a priori would then just be subtracting a constant. We added this figure here for the referee just to be clear, for 1 day in March 2011 :

[Figure]

P5,L31: From Google maps I conclude that these 3 locations are all over the sea – but it would be helpful to state that in the text or the figure caption. If you ha*ve only three examples, I don't think it is useful to present both northern and southern mid-latitudes, which one would expect to be similar (particularly near the equinox). It would be more informative to have different land-air temperature contrasts instead, representing the max/min values shown in Fig 6.*

Indeed, they all seem to be over the sea. We have created these figures randomly and want to give  the reader a quick overview on the shape of the averaging kernel, at different geographical locations. Of course, these are 'just 3 examples' but they are representative of the general conditions we get at mid-latitudes and in the tropics. We believe they are more informative than giving the averaging kernels at the max/min thermal contrast values. Figure 7 gives a more elaborate overview of the whole range of sensitivity of the IASI CH4 measurements.

We regenerated a few figures and replaced the 2 figures on the left to have not just 3 locations over the sea. The averaging kernels are very much the same.

We changed the caption of Figure 5 accordingly:
CH4 averaging kernels for 3 pixels on the 1st of March 2013 at 3 different locations (55°N, 4°N and 47°S)
$\rightarrow$
CH4 averaging kernels for 3 pixels on the 1st of March 2013 at 3 different locations (52°N, 4°N and 47°S)

In the text we changed :
P5,L31:
..at 3 different geographical locations; at northern mid-latitudes (55°N), in the tropics..
$\rightarrow$
..at 3 different geographical locations; at northern mid-latitudes (52°N), in the tropics..

P6,L4:
One independent piece of information (1.01 < DOFS < 1.47)..
$\rightarrow$
One independent piece of information (1.01 < DOFS < 1.45)..

[Figure]

**Figure 5.** CH₄ averaging kernels for 3 pixels on the 1$^{st}$ of March 2013 at 3 different locations (52°N, 4°N and 47°S).

*P6,L10: 'thermal contrast' needs to be defined.*
We changed the sentence (P6, L10):
The variability of the AK (and hence the DOFS) is dependent on the thermal contrast..
→
The variability of the AK and hence the DOFS is dependent on the thermal contrast (the difference between the surface temperature and the temperature of the first atmospheric vertical layer)..

*P6,L20: 'one independent piece of information is retrieved with good sensitivity'. While DOFS ~ 1, it would be more useful to have some idea of, for example, how this translates to a reduction in the a priori uncertainty for a 4-17km column.*
Section 4.1 on Information content explains the vertical sensitivity of the retrieved profile. We believe it's important that the reader of the manuscript understands that although a profile of CH4 is retrieved there is only one piece of information, which is what we would like to stress in this chapter. We therefore like to keep this section as it is, with the information as is given.

*P6,L23: What limits the latitude coverage? Here it says 60S-70N but two of the NDACC comparison sites are higher than 70N.*
P6, L23: The limits were chosen to produce the necessary dataset for the GHG-CCI project. For the NDACC validation the CH4 profiles were retrieved at the different NDACC station locations and consequently at two NDACC sites at latitudes higher than 70°N.

*P6,L23: 'binned on a 1x1 deg grid'. So does this mean the plotted points represent not just an average of the four pixels but an average of all the pixels within the 1x1 box? Or is the binning some other process? And what happens if, say one of the 4 pixels is flagged as cloud-contaminated. Is the average then made of the remaining 3 or is this set of 4 pixels excluded?*
It is the average of all the pixels within the 1x1 degree box. If one of the 4 pixels is flagged as cloud-contaminated the set of 4 pixels is excluded.

*P8,L24 gives IASI a priori systematic component as 2% of a priori value. Where does this come from? Just the error in WACCM? No systematic component of IASI retrieval uncertainty is considered.*

P8, L24: The 2% a priori systematic component is an estimate that the a priori values at certain locations could be under- or overestimated. We assume 1 single a priori profile for the whole globe whose values could easily be underestimated at certain locations.

We double checked all the references and added the bibliography now with a bibtex file.

Fig 1: Rather than just show a generic piece of spectrum it would be helpful if this figure was also used to show the individual contributions of different molecules to this spectral region (eg separate panel with same x-axis).
We added a separate panel with the contributions of the different molecules, see the figure below.

We changed the caption of figure 2 accordingly:
[top] Measured (blue) and simulated (yellow) radiances. [bottom] Measured minus simulated radiances. The mean difference (bias), 1-$\sigma$ standard deviation of the difference and radiometric noise-value used in the retrieval (all in x $10^{-8}$ W/(cm$^2$ sr cm$^{-1}$)) are given in the title.
$\rightarrow$
Top panel : [top] Measured (blue) and simulated (yellow) radiances. [bottom] Measured minus simulated radiances. The mean difference (bias), 1-$\sigma$ standard deviation of the difference and radiometric noise-value used in the retrieval (all in x $10^{-8}$ W/(cm$^2$ sr cm$^{-1}$)) are given in the title. Bottom panel : Three simulated radiances under the assumption of a single-species atmosphere containing either only CH4, H2O or N2O, showing the contribution of the different prominent molecules in this spectral region.

And added the following sentences in the text on P4, L12 :
In the lower panel, the overlapping contributions of the different molecules CH4, H2O and N2O are illustrated. Here the radiances are simulated under the assumption of a single-species atmosphere containing either CH4, H2O or N2O. The top panel shows a negligible bias and..

[Figure]

Fig 2: On the left panel it would be helpful to also have the a priori error bars plotted for comparison.
We added a shaded area in Fig. 2 which represents the a priori variability as calculated from the square root of the diagonal of the a priori covariance matrix. We added the following sentence to the caption of Fig.2:
The pink shaded area is the a priori variability and the horizontal blue bars are the retrieval uncertainty.

[Figure]

**Figure 2.** [left] Retrieved and a priori CH4 vmr profile in ppmv for an observation on the 2[nd] of July 2013. The pink shaded area is the a priori variability and the horizontal blue bars are the retrieval uncertainty. [middle] Averaging kernel of the retrieval with a DOFS of 1.40. [right] CH4 uncertainty profiles in percentage. Given are the measurement (yellow) and smoothing (blue) uncertainty which contribute to the total (purple) uncertainty. The black line represents the variability of the a priori as calculated from the square root of the diagonal elements of the a priori uncertainty covariance matrix $S_a$.

*Table 1: This lists a priori information as WACCM, but that is only for certain elements of the state vector. 'IASI L2' should include the word 'EUMETSAT' for consistency with the text. There should also be something about the a priori covariance information.*
Table 1: IASI L2 has been added to the a priori information. The description in the text (Sect 3.2) regarding the a priori covariance information is in my opinion sufficient.

*Table 2: I'm surprised at the spread in systematic errors in Table 2 for the various NDACC comparisons. Assuming this represents a combination of the NDACC systematic error budget and the 2% systematic error assumed for the retrieval a priori, this variation must mostly come from the NDACC data. Yet Sepulveda et al (2014) quotes a figure of 2.5% which is largely spectroscopic uncertainty (and therefore common to all sites). And the fact that these systematic errors are all much larger than the biases suggests something wrong.*
The spread in systematic uncertainties in Table 2 is indeed due to the variation in systematic uncertainty estimates of the NDACC dataset which is not consistent for the different NDACC stations. As mentioned in the text, the NDACC CH4 retrieval is not fully harmonized yet. The implementation of a fully harmonized retrieval is ongoing work as part of the GAIA-CLIM project. A harmonization of the uncertainty estimates is also part of that work. Sepulveda et al. (2014) did a separate study where they analyzed the spectra at 9 NDACC stations with a different inversion code PROFFIT (the NDACC FTIR community uses the SFIT software) and they did a separate error analysis. This is not the same data that is publicly available on the NDACC data site.

We added the following 2 sentences on P8, L33:
They were found to be less than 1% for 6 out of the 10 NDACC stations. As mentioned before, a full harmonization of the NDACC CH4 retrieval is ongoing..
$\rightarrow$
They were found to be less than 1% for 6 out of the 10 NDACC stations. Also note the spread in uncertainty estimates, especially for the systematic component. This is due to the differences in

reported systematic and random error covariances from the different NDACC stations. As mentioned before, a full harmonization of the NDACC CH4 retrieval is ongoing..

TYPOGRAPHICAL ERRORS/SUGGESTIONS
We implemented the corrections the referee suggested. Thank you for being so observant.

P1,L3: usefullness → usefulness

P2,L2: greenhouse gas-intensive →  greenhouse gas-intensive

P2,L3 + P2,L7 +  P2,L9 + P9,20 + P19, caption Figure 4: ppb → ppbv

P2,L5 + P16,L6 :WMO News Bulletin, 2016 → WMO News Bulletin, 2016).

P2,L31: LIDAR → lidar

P3,L6:
The Infrared Atmospheric Sounding Interferometer (IASI) onboard MetOp, is a thermal cross-nadir scanning infrared sounder.
→ The Infrared Atmospheric Sounding Interferometer (IASI) onboard MetOp is a thermal cross-nadir scanning infrared sounder.

P3,L7:
Launched in October 2006, it is the first in a successive series of three..
→ Launched in October 2006, it is the first in a series of three..

P3,L16:
..for numerical weather predictions, the IASI mission..
→ ..for numerical weather prediction, the IASI mission..

P3,L19: On the longer term  the continuity of the program is ensured.. → In the longer term the continuity of the programme is ensured..

P3,L20:  ..the IASI observation.. →  ..the IASI observations..

P3,L23: ASIMUT-ALVL is a modular software for radiative transfer (RT) calculations.. → ASIMUT-ALVL is modular software for radiative transfer (RT) calculations..

P4,L3: ..instrument on-board ExoMars.. → ..instrument onboard ExoMars..

P4,L8:
The spectral range considered for the CH4 retrieval is the 1210-1290 cm−1 range covering part of the ν4 spectral band. →
The spectral range considered for the CH4 retrieval is the 1210-1290 cm−1 region covering part of the ν4 spectral band.

P4,L12: ..and its residual.. → ..and the residual..

P4,L14: Certain spectral ranges in the considered spectral band.. → Certain spectral ranges in the considered spectral region...

P4,L22: ..are summarized in Table 6. →  ..are summarized in Table 61.

P4,L28: ..its a priori.. → ..their a priori..

P4,L32: water vapor → water vapour

P5,L3: diplays → displays

P4,L5: .. constrained with of an a priori.. → ..constrained with an a priori..

P4,L11: su rface → surface

P5,L17: ..10 Tb which is reduced to 1 Tb/year.. →  ..10 Tb which is reduced to 1 Tb..

P5,L19: 4x10−9 W/(m2 sr m−1 )) → 4x10−9 W/(cm2 sr cm−1 ))

P5, L20: 2x10−8 W/(m2 sr m−1 )) → 2x10−8 W/(cm2 sr cm−1 ))

P5,L24: negligibles → negligible

P6,L30: van Weele et al. (2011) examined.. → Van Weele et al. (2011) examined..

P7,L17: ..with less than 200 collocations..  →  ..with fewer than 200 collocations..

P8,L5: alitude → altitude

P8,L30: slighlty → slightly

P12,L32: Forc- ing → Forcing

Fig 1: insert space: 'sr cm'
This has been implemented.

**CHANGES VALIDATION RESULTS:**

**ABSTRACT**

IASI CH4 partial columns are found to correlate well with the ground-based data for 7 out of the 10 Fourier Transform Infrared (FTIR) stations with correlation coefficients between 0.71 and 0.96. Mean differences between IASI and FTIR CH4 range between -1.93 and 4.40% and are within the systematic uncertainty. For 7 out of the 10 stations absolute differences are less than 1%. The standard deviation of the difference lies between 1.40 and 3.99% for all the stations.

→

IASI CH4 partial columns are found to correlate well with the ground-based data for 6 out of the 10 Fourier Transform Infrared (FTIR) stations with correlation coefficients between 0.60 and 0.84. Relative mean differences between IASI and FTIR CH4 range between -2.31 and 4.04% and are within the systematic uncertainty. For 6 out of the 10 stations the relative mean differences are smaller than ±1%. The standard deviation of the difference lies between 1.76 to 2.97% for all the stations.

**SECT. 5 VALIDATION**

P8, L5:

.. where the FTIR measurement has maximum sensitivity (typically at 5km altitude on the LOS).

→

.. where the FTIR measurement has maximum sensitivity (typically at 5km altitude on the LOS). To guarantee a certain homogeneity of the NDACC data with NDACC CH4 profiles of comparable quality we applied a filtering on some of the NDACC data when large outliers where found. We also applied a filtering to the IASI CH4 profiles. We omitted IASI pixels with DOFS < 0.85 and when the root mean square of the residual $> 2.2 \times 10^{-8}$ W/(cm 2 sr cm $-1$ ).

P8, L9:

Relative mean differences between IASI and NDACC lie between -1.93 and 0.67% (of which 7 stations out of 10 less than ±1%) with exception of the Thule station, where IASI is biased high with respect to NDACC by 4.4%. The standard deviation of the difference lies in the range 1.91 to 3.99% for the 10 stations.

→

Relative mean differences between IASI and NDACC lie between -2.31 and 0.18% (of which 6 stations out of 10 less than ±1%) with exception of the Thule station, where IASI is biased high with respect to NDACC by 4.04%. The standard deviation of the difference lies in the range 1.76 to 2.97% for the 10 stations.

P8, L29:

Also for Thule the mean difference of 4.40% is within the systematic uncertainty of 5.14%.

→

Also for Thule the mean difference of 4.04% is within the systematic uncertainty of 5.28%.

P8, L30:

The standard deviation of the difference is close to the random uncertainty, but for certain stations slightly larger. This discrepancy could be due to an additional error associated with the grid conversions or a geolocation error which are not taken into account (Calisesi et al., 2005; Vigouroux et al., 2009). Another reason could be the current underestimation of the random uncertainty of the NDACC CH 4 retrievals. They were found to be less than 1% for 6 out of the 10 NDACC stations. The ongoing work in the GAIA-CLIM project will harmonize the error characterization for all NDACC stations in the coming period. This comparison stresses the importance of this harmonization work.

→
The standard deviation of the difference is within the random uncertainty for all stations. We did notice a current underestimation of the random uncertainty of the NDACC CH4 retrievals. They were found to be less than 1% for 6 out of the 10 NDACC stations. Also note the spread in uncertainty estimates, especially for the systematic component. This is due to the differences in reported systematic and random error covariances from the different NDACC stations. The ongoing work in the GAIA-CLIM project will harmonize the error characterization for all NDACC stations in the coming period. This comparison stresses the importance of this harmonization work.

P9, L2:
Scatter plots of collocated partial columns are presented in Fig. 10. We find very good correlations (R=0.71-0.88) for the high-latitude stations Eureka, Thule and Kiruna. High correlations (R>0.9) are found as well for the mid-latitude stations Jungfraujoch and Zugspitze, while the mid-latitude stations Toronto performs poorer with a correlation of 0.44. The tropical island stations Maido and Mauna Loa show poor correlations (R=0.41-0.50) although biases are below 1% for these stations. The tropical island station Izaña however shows a high correlation of 0.90. For the most Southern station Wollongong (34◦S) we find a good correlation of 0.77. The poorer correlations at the Mauna Loa, Maido and Toronto stations could be attributed to the lower CH4 variability we see at these locations compared to the other stations. In addition, at Mauna Loa and Toronto, we see a few outliers which could explain the poorer linear regression fit at these stations. These results demonstrate the ability of the IASI data to well capture the temporal variation of CH4 .
→
Scatter plots of collocated partial columns are presented in Fig. 10. We find good correlations (R=0.67-0.84) for the high-latitude stations Eureka, Thule and Kiruna. Good correlations are found as well for the mid-latitude stations Jungfraujoch (R=0.81) and Zugspitze (R=0.68), while the mid-latitude station Toronto performs poorer with a correlation of 0.52. The tropical island stations Izaña, Maido and Mauna Loa show poor correlations (R=0.15-0.36) although biases are below 1.20% for these stations. For the most Southern station Wollongong (34◦S) we find a correlation of 0.60. Several tests were performed to explain the poorer correlations found at the tropical island stations. We applied a stronger filtering on the IASI and NDACC data but found no improvement. We investigated a possible relation of IASI land or IASI sea pixels with differences between the IASI and NDACC retrieved CH4 but found no correlation. We therefore attribute the poorer correlations at the Izaña, Mauna Loa and Maido stations to the lower CH4 variability we see at these locations compared to the other stations. In addition, at Maido and Mauna Loa, we see a few outliers which could explain the poorer linear regression fit at these stations.

**SECT. 6 DISCUSSION, CONCLUSION AND OUTLOOK**
P9, L25 :

[revised manuscript text omitted]